

# Exact spin correlators of integrable quantum circuits from algebraic geometry

Arthur Hutsalyuk[1,2], Yunfeng Jiang[3*], Balazs Pozsgay[2], Hefeng Xu[4] and Yang Zhang[4,5]

**1** SISSA and INFN, Sezione di Trieste, via Bonomea 265, 34136 Trieste, Italy
**2** MTA-ELTE "Momentum" Integrable Quantum Dynamics Research Group,
Department of Theoretical Physics, ELTE Eötvös Loránd University,
Pázmány Péter stny. 1A, Budapest 1117, Hungary
**3** School of Physics and Shing-Tung Yau Center, Southeast University,
Nanjing 210096, Jiangsu, China
**4** Interdisciplinary Center for Theoretical Study,
University of Science and Technology of China, Hefei, Anhui 230026, China
**5** Peng Huanwu Center for Fundamental Theory, Hefei, Anhui 230026, China

* jinagyf2008@seu.edu.cn

## Abstract

We calculate the correlation functions of strings of spin operators for integrable quantum circuits exactly. These observables can be used for the calibration of quantum simulation platforms. We use the algebraic Bethe Ansatz, in combination with computational algebraic geometry, to obtain analytic results for medium-size (around 10-20 qubits) quantum circuits. The results are rational functions of the quantum circuit parameters. We obtain analytic results for such correlation functions both in the real space and Fourier space. In the real space, we analyze the short time and long time limits of the correlation functions. In Fourier space, we obtain analytic results in different parameter regimes, which exhibit qualitatively different behaviors. Using these analytic results, one can easily generate numerical data to arbitrary precision.



# 1 Introduction

Integrable models are special many-body systems, which allow for the exact computation of physical observables in various situations [1, 2]. These exact results can then be compared to measurements in real-world experiments. For many decades, research has focused on the physical quantities in equilibrium, and in many cases, spectacular agreement was found between theory and experiment. Examples include the measurement of the exact free energy of the 1D Bose gas [3], the computation of static correlation functions of the Heisenberg spin chain [4], and the experimental observation of the $E_8$-related spectrum of the perturbed Ising model [5–9].

However, in the last decade, the focus of the research shifted towards non-equilibrium situations. This was motivated by advances on the experimental side: it became possible to realize well-known integrable models in various types of experiments, enabling the study of non-equilibrium dynamics in these systems. Examples of experiments with ultra-cold atoms include the famous "Quantum Newton's cradle" experiment [10], the observation of the Generalized Gibbs Ensemble [11], and the confirmation of Generalized Hydrodynamics [12].

As an alternative to experiments with ultra-cold gases, it has also become possible to simulate integrable models on quantum computers, as demonstrated in the recent experiment of Google Quantum AI [13]. In a quantum computer, the time evolution is performed in discrete steps, via the succession of local operations. It is possible to perform these steps in a homogeneous manner, thus obtaining a so-called brickwork quantum circuit. Such circuits can be integrable, and in such cases they are often related to well-known statistical physical models and quantum spin chains studied earlier in the literature [14, 15]

The integrable circuit studied in [13] is related to the six-vertex model and the Heisenberg spin chain [16]. Even though these models are well known, relatively few exact results have been published for the corresponding brickwork circuits; for example, see [14, 17]. The Bethe Ansatz solution of this specific circuit was considered in [18], where it was argued that a certain correlation function of this circuit is a measurable quantity in experiments. These correlation functions are crucial for error mitigation for quantum simulation platforms [19]. However, the explicit computation of this correlation function was not given in [18], except for eigenstates with only two magnons.

In this work we revisit the problem of [18] and we compute closed form results for the correlation function proposed there. The correlation functions in question can be computed using the standard form factor expansion; however, such computation will involve summation over all eigenstates of the finite volume system with given global quantum numbers. Therefore, any theoretical method to compute these objects has to treat all eigenstates, and in Bethe Ansatz this means finding all physical solutions to the Bethe equations. This is a notoriously difficult task, see for example [20]. Instead of solving the Bethe Ansatz equations numerically, we apply recent developments in integrability, including the rational $Q$-system [21–29] and algebro-geometric approach [30–34], which allows us to calculate the final result *analytically*. We also make use of the algebraic Bethe Ansatz, which allows us to use the full power of integrability in the computation of correlation functions and obtain results beyond two-magnon states.

Before delving into the technical details, we first present the main results and implications of this work.

**Exact results** First, we obtain exact results for correlation functions of an integrable quantum circuit in both real space (in section 4) and momentum space (in section 5 and appendices E and F). These results are valuable for comparison with experimentally measured quantities, particularly in the context of error mitigation. A key advantage of our findings is that they are both exact and analytical, requiring no approximations, and can, in principle, be evaluated with arbitrary precision. We have computed these results for systems ranging from 10 to 20 qubits, which correspond to the scales of real-world experiments. The specific parameter values chosen are not critical; they were selected for illustrative purposes and can be easily adjusted to other values in the calculations.

**The method** Second, the method employed in this work demonstrates that by integrating the rational $Q$-system approach with computational algebraic geometry, we can compute a range of physically relevant quantities for medium-sized systems, such as the correlation functions examined here. This work extends previous results on the computation of partition functions for the six-vertex model [31, 32, 34]. The method itself holds significant potential and can be applied to compute other quantities of interest for quantum integrable models solvable via the Bethe Ansatz.

**Lee-Yang zeros** Finally, by analyzing our results in various limits, we observe several intriguing phenomena. One key highlight is the zeros of the correlation function. The correlation

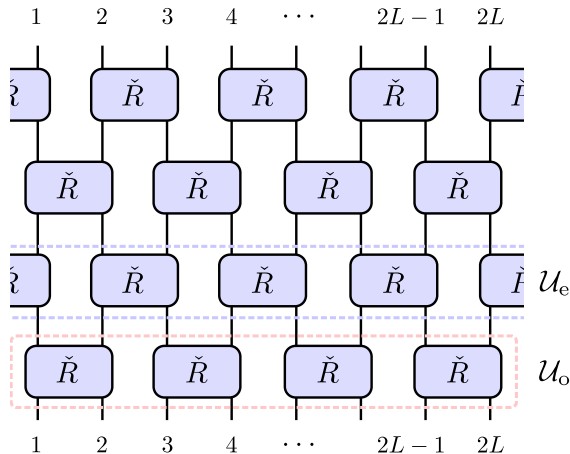

Figure 1: Graphical depiction of the quantum circuit in question. The vertical direction is associated with the direction of time. Each rectangle stands for the action of the unitary two-site gate $\check{R}$, given in the main text.

function computed in this work is the lattice analog of the Loschmidt amplitude, a quantity that plays a pivotal role in dynamical quantum phase transitions [35–37]. The Lee-Yang zeros for the Loschmidt echo are of great interest in characterizing different dynamical quantum phases. In the long-time limit, we find that the zeros of the correlation functions condense onto specific curves, which separate distinct phases. From this perspective, our results represent the first step toward modeling dynamical quantum phase transitions on quantum circuits and provide the first theoretical prediction for the Lee-Yang zeros of the Loschmidt echo.

The rest of the paper is structured as follows. In section 2, we introduce the integrable quantum circuit model and the quantities that we would like to compute. In section 3, we introduce the main tools for computations, namely algebraic Bethe Ansatz and computational algebraic geometry. The results are presented in the following two sections. In section 4, we present results in real space. We analyze the 'short time' limit and 'long time' limit where the discrete 'time' here refers to the number of layers in the quantum circuit. We have found that for short times the result is independent of the size of the system if the system size is sufficiently large. In section 5 we present the exact results in Fourier space. We computed the result explicitly for two sets of parameters of the quantum circuit. They exhibit qualitatively different behaviors. We conclude in section 6 and discuss future directions. More technical details are delegated to the appendices.

## 2 The model and the observables

In this section, we introduce the model that we are going to study and the physical quantities that we want to compute.

### 2.1 The model

We are interested in the dynamics of a one-dimensional quantum cellular automaton, where the time evolution is given by the successive application of local two-site gates. The arrangement of these gates is depicted in Figure 1; such a system is often called a brickwork quantum circuit, and it can be regarded as a special type of a Floquet system. For a formal definition of the circuit, see below.

**Evolution operator**   The operator which evolves the whole system in time (depicted as the vertical direction in Figure 1) is defined as

$$\mathcal{U} = \mathcal{U}_{\mathrm{e}} \mathcal{U}_{\mathrm{o}} \, , \tag{1}$$

where

$$\begin{aligned}
\mathcal{U}_{\mathrm{o}} &= U_{12} U_{34} \cdots U_{2L-1,2L} \, , \\
\mathcal{U}_{\mathrm{e}} &= U_{23} U_{45} \cdots U_{2L,1} \, ,
\end{aligned} \tag{2}$$

and $U_{j,j+1}$ stands for a two-site unitary operator (acting on sites $j$ and $j+1$) to be specified below.

**Two-site unitary**   The operator $U$ that we consider was given in [18], and it reads

$$U = \begin{pmatrix} 1 & 0 & 0 & 0 \\ 0 & \cos\alpha & \mathrm{i}\sin\alpha & 0 \\ 0 & \mathrm{i}\sin\alpha & \cos\alpha & 0 \\ 0 & 0 & 0 & e^{2\mathrm{i}\phi} \end{pmatrix} . \tag{3}$$

Here $\alpha$ and $\phi$ are real numbers, such that $U$ is unitary. In the spin chain language, this operator conserves the $z$ component of the spin,[1] which implies that the time evolution operator $\mathcal{U}$ will commute with the global $S^z$ operator given by

$$S_z = \frac{1}{2}(\sigma_1^z + \ldots + \sigma_{2L}^z) \, . \tag{4}$$

We will show that by a proper change of variables, this unitary gate is equivalent to a certain evaluation of the so-called $\check{R}$-matrix known in the integrability literature. To make this identification, we first factor out an operator built from $\sigma^z$ and a phase factor

$$U = e^{\mathrm{i}\phi} e^{-\mathrm{i}\phi(\sigma_1^z + \sigma_2^z)/2} \begin{pmatrix} 1 & 0 & 0 & 0 \\ 0 & e^{-\mathrm{i}\phi}\cos\alpha & \mathrm{i}e^{-\mathrm{i}\phi}\sin\alpha & 0 \\ 0 & \mathrm{i}e^{-\mathrm{i}\phi}\sin\alpha & e^{-\mathrm{i}\phi}\cos\alpha & 0 \\ 0 & 0 & 0 & 1 \end{pmatrix} . \tag{5}$$

After multiplying all the $U$ operators, the factors in front simply yield a global phase $e^{-\mathrm{i}\phi(2S^z+L)}$. The so-called $\check{R}$-matrix of the XXZ spin chain is written in the traditional notation as

$$\check{R}(u) = \begin{pmatrix} 1 & 0 & 0 & 0 \\ 0 & c(u) & b(u) & 0 \\ 0 & b(u) & c(u) & 0 \\ 0 & 0 & 0 & 1 \end{pmatrix} . \tag{6}$$

The matrix elements of (6) read

$$b(u) = \frac{\varphi(u)}{\varphi(u+\eta)} \, , \qquad c(u) = \frac{\varphi(\eta)}{\varphi(u+\eta)} \, , \tag{7}$$

where $\varphi(u) = \sinh u$ and $\eta$ are related to the anisotropy parameter $\Delta$ of the XXZ chain via $\Delta = \cosh\eta$. It is useful to consider also the so-called rational limit corresponding to the $SU(2)$-symmetric XXX chain, in which case $\varphi(u) = u$ and $\eta = \mathrm{i}$.

---

[1] In what follows we will often use the language of spin chains. Therefore we shall use qubits and spins interchangeably.

Now we can read off the identification between the two different parameterizations given in (5) and (6). For the generic case, we find

$$e^{-i\phi}\cos\alpha = \frac{\sinh\eta}{\sinh(u+\eta)}, \qquad ie^{-i\phi}\sin\alpha = \frac{\sinh u}{\sinh(u+\eta)}. \tag{8}$$

We can easily solve for $\alpha$ and $\phi$ in terms of $u$ and $\eta$ as follows:

$$\tan\alpha = -i\frac{\sinh u}{\sinh\eta}, \qquad e^{-2i\phi} = -\frac{\sinh(u-\eta)}{\sinh(u+\eta)}. \tag{9}$$

The rational limit is obtained only when $\alpha = \phi$.

In [18], the parameters $\alpha$ and $\phi$ are real numbers, but for our intermediate computations we can work with generic complex $u$ and $\eta$. In this parametrization, $u$ has the interpretation of the spectral parameter.

From now on, we shall work with the $u, \eta$ parametrization and write the two-site unitary as $\check{R}_{ij}(u)$ as in the integrability literature where $i$ and $j$ denote the sites that it acts on.

The most important property of $\check{R}_{ij}$ is that it satisfies the quantum Yang-Baxter equation

$$\check{R}_{ij}(u)\check{R}_{ik}(u+v)\check{R}_{jk}(v) = \check{R}_{jk}(v)\check{R}_{ik}(u+v)\check{R}_{ij}(u), \tag{10}$$

which underlies the integrability of the model. In the current situation, the indices are always adjacent, *i.e.* we can take $j = i+1$, $k = i+2$ in the above equation.

**Remarks** This model is well known in the literature of integrable models, and it made its appearance in several different contexts, which we briefly summarize below.

1. It first appeared in integrable lattice models, which describe classical 2D lattice systems, by positioning the 6-vertex model on a diagonal lattice [16]. In this context, $\check{R}_{ij}$ encodes the local Boltzmann weights and $\mathcal{U}$ is known as the diagonal-to-diagonal transfer matrix, which can be used to compute lattice partition functions.

2. The above model also serve as a lattice discretization of the quantum sine-Gordon model, which was used by Destri and de Vega to compute the finite size energy spectrum [38–40]. In this context, $\mathcal{U}$ is related to the evolution in the light cone directions.

3. The model appeared in [14] as the integrable Trotterization of the XXZ spin chain and was further studied in [15].

We want to emphasize that although the underlying model is equivalent, the contexts are quite different. The interesting physical quantities depend on the context. We will now turn to the quantities that we are interested in in the current context.

## 2.2 The physical quantity

In statistical mechanics and condensed matter physics, research is often focused on the properties of the ground states or the first few low-lying excited states. These computations are typically done in the thermodynamic limit. However, in the current context, we are considering a quantum circuit with finite qubits and discrete time evolution. Therefore, there is no notion of a ground state. Furthermore, we will not consider the thermodynamic limit because we intend to compute exact results with a fixed finite number of qubits or spins.

The quantity that can be measured conveniently for quantum computers are correlation functions of local spin operators acting on the pseudovacuum state [18]. Let us denote the basis states of each qubit by $|0\rangle$ and $|1\rangle$. The pseudovacuum state is defined by

$$|\Omega\rangle = |0\rangle_1 \otimes |0\rangle_2 \otimes \cdots \otimes |0\rangle_{2L} \equiv |00\cdots 0\rangle. \tag{11}$$

We consider the following string of local spin operators

$$\hat{\Sigma}_k^{(N)} = \sigma_{k+1}^x \sigma_{k+2}^x \cdots \sigma_{k+N}^x, \tag{12}$$

which acts on $N$ sites starting from site $k$.

Acting on the pseudovacuum, this operator creates a domain wall state. For concreteness, let us define

$$|\text{DW}_N\rangle \equiv \hat{\Sigma}_0^{(N)}|\Omega\rangle = |\underbrace{11\cdots 1}_{N}0\cdots 0\rangle. \tag{13}$$

We are interested in the dynamical correlation function

$$\langle\Omega|\hat{\Sigma}_k^{(N)}(0)\hat{\Sigma}_0^{(N)}(n)|\Omega\rangle, \tag{14}$$

where time evolution of the operators is understood in the Heisenberg picture:

$$\hat{\Sigma}_0^{(N)}(n) \equiv U^n \hat{\Sigma}_0^{(N)}(0) U^{-n}. \tag{15}$$

Apart from trivial phases, the pseudovacuum is a fixed point of the time evolution operator; therefore, the correlation can also be expressed as

$$\mathcal{D}^{(N)}(n,k) = \langle\Omega|\hat{\Sigma}_k^{(N)} \mathcal{U}^n \hat{\Sigma}_0^{(N)}|\Omega\rangle. \tag{16}$$

For $k = 0$, this quantity is expressed further as

$$\langle\text{DW}_N|\mathcal{U}^n|\text{DW}_N\rangle, \tag{17}$$

which is the discrete version of the so-called Loschmidt echo or return amplitude (fidelity) in quantum quench dynamics.

It is also useful to consider the Fourier transform of $\mathcal{D}^{(N)}$ defined by[2]

$$\widehat{\mathcal{D}}^{(N)}(\omega, Q) = \sum_{n=0}^{\infty} e^{-n(i\omega + \gamma)} \sum_{k=1}^{L} e^{iQk} \mathcal{D}^{(N)}(n, 2k), \tag{18}$$

where the real parameter $0 < \gamma \ll 1$ is introduced to provide proper convergence for the time Fourier transform. Here $Q$ is called the quasi-momentum and it satisfies the quantization condition $e^{iQL} = 1$.

Our goal is to compute the correlation functions (16) and (18) analytically for given quantum numbers.

## 3 Bethe Ansatz and algebraic geometry

Analytic computation of quantities like (16) and (18) are usually quite hard. Thanks to the integrability of the model, we can obtain analytic results for these quantities using Bethe Ansatz and computational algebraic geometry. In this section, we discuss how to compute observables of the integrable quantum circuits using these powerful theoretical tools.

---

[2]Notice that in the summand we have $\mathcal{D}^{(N)}(n, 2k)$ instead of $\mathcal{D}^{(N)}(n, k)$. This definition is more natural because the system is invariant under a 2-site translation.

## 3.1 Algebraic Bethe Ansatz

**Transfer matrices**  In order to compute the time evolution by $\mathcal{U}$ analytically, we diagonalize it by Bethe Ansatz and work with its eigenvalues. In the integrability literature, $\mathcal{U}$ is identified with the so-called diagonal-to-diagonal transfer matrix. Its explicit diagonalization is achieved by a well-known relation between the diagonal-to-diagonal transfer matrix and the row-to-row transfer matrix, which we will briefly recall now. We start with the $R$-matrix of the XXZ spin chain

$$R_{ij}(u) = \begin{pmatrix} 1 & 0 & 0 & 0 \\ 0 & b(u) & c(u) & 0 \\ 0 & c(u) & b(u) & 0 \\ 0 & 0 & 0 & 1 \end{pmatrix}_{ij}. \tag{19}$$

This $R$-matrix is related to the $\check{R}$-matrix by $\check{R}_{ij}(u) = P_{ij}R_{ij}(u)$ where $P_{ij}$ is the permutation operator

$$P_{ij} = \begin{pmatrix} 1 & 0 & 0 & 0 \\ 0 & 0 & 1 & 0 \\ 0 & 1 & 0 & 0 \\ 0 & 0 & 0 & 1 \end{pmatrix}_{ij}. \tag{20}$$

Following the standard procedure of the algebraic Bethe Ansatz (see *e.g* [41]), we can define the following monodromy matrices

$$M_0(u|\{\theta_j\}) = R_{01}(u - \theta_1)R_{02}(u - \theta_2)\cdots R_{0,2L}(u - \theta_{2L}), \tag{21}$$
$$\widehat{M}_0(u|\{\theta_j\}) = R_{0,2L}(u + \theta_{2L})\cdots R_{0,2}(u + \theta_2)R_{0,1}(u + \theta_1),$$

where '0' denotes an auxiliary space and we have introduced a set of parameters $\{\theta_j\}$ called the inhomogeneities. The row-to-row transfer matrix is defined as

$$\mathcal{T}(u|\{\theta_j\}) = \text{tr}_0 M_0(u|\{\theta_j\}), \qquad \widehat{\mathcal{T}}(u|\{\theta_j\}) = \text{tr}_0 \widehat{M}_0(u|\{\theta_j\}). \tag{22}$$

Our evolution operator $\mathcal{U}$ defined in (1), (2) is related to the row-to-row transfer matrix (22) by [42]

$$\mathcal{U}(u) = \mathcal{T}\left(\tfrac{u}{2}|\theta_j(\tfrac{u}{2})\right)\widehat{\mathcal{T}}\left(\tfrac{u}{2}|\theta_j(\tfrac{u}{2})\right), \tag{23}$$

where $\theta_j(\tfrac{u}{2})$ stands for the following choice of the inhomogeneities

$$\theta_j(\tfrac{u}{2}) = \left\{-\tfrac{u}{2}, \tfrac{u}{2}, \ldots, -\tfrac{u}{2}, \tfrac{u}{2}\right\}. \tag{24}$$

The relation (23) can be proven by using the so-called regularity condition $R_{ij}(0) = P_{ij}$, which is valid for our $R$-matrix (19).

The transfer matrix $\widehat{\mathcal{T}}$ is related to the transfer matrix $\mathcal{T}$ by

$$\mathcal{T}(-u - \eta|\{\theta_j\}) = \widehat{\mathcal{T}}(u|\{\theta_j\}), \tag{25}$$

which can be proven straightforwardly by using the crossing relation of the $R$-matrix

$$R_{12}(-u - \eta) = -\sigma_1^y R_{12}^{t_1}(u)\sigma_1^y. \tag{26}$$

The row-to-row transfer matrix $\mathcal{T}$ can be diagonalized by Bethe Ansatz. We denote each eigenstate of $\mathcal{T}$ by $|\mathbf{u}_K\rangle$ where $\mathbf{u}_K = \{u_1, u_2, \ldots, u_K\}$ are the Bethe roots that are rapidities of magnons. We have

$$\mathcal{T}(u|\{\theta_j\})|\mathbf{u}_K\rangle = t(u|\{\theta_j\})|\mathbf{u}_K\rangle, \tag{27}$$

where the eigenvalue is given by

$$t(u|\{\theta_j\}) = \prod_{k=1}^{K} \frac{\varphi(u-u_k-\eta)}{\varphi(u-u_k)} + \prod_{j=1}^{2L} \frac{\varphi(u-\theta_j)}{\varphi(u-\theta_j+\eta)} \prod_{k=1}^{K} \frac{\varphi(u-u_k+\eta)}{\varphi(u-u_k)} \,. \tag{28}$$

The Bethe roots satisfy Bethe Ansatz equations (BAE)

$$\prod_{a=1}^{2L} \frac{\varphi(u_k-\theta_a+\eta)}{\varphi(u_k-\theta_a)} = \prod_{j\neq k}^{K} \frac{\varphi(u_k-u_j+\eta)}{\varphi(u_k-u_j-\eta)} \,, \qquad k=1,\dots,K \,. \tag{29}$$

Using the relation (23) and (25), we obtain the eigenvalue of $\mathcal{U}$

$$\mathcal{U}|\mathbf{u}_K\rangle = \tau(\mathbf{u}_K)|\mathbf{u}_K\rangle \,, \tag{30}$$

where

$$\tau(\mathbf{u}_K) = \prod_{k=1}^{K} \frac{\varphi(u_k-\frac{u}{2}+\eta)}{\varphi(u_k-\frac{u}{2})} \frac{\varphi(u_k+\frac{u}{2})}{\varphi(u_k+\frac{u}{2}+\eta)} \,. \tag{31}$$

For the choice of inhomogeneities (24), the BAE (29) becomes

$$\left( \frac{\varphi(u_k-\frac{u}{2}+\eta)}{\varphi(u_k-\frac{u}{2})} \right)^L \left( \frac{\varphi(u_k+\frac{u}{2}+\eta)}{\varphi(u_k+\frac{u}{2})} \right)^L = \prod_{j\neq k}^{K} \frac{\varphi(u_k-u_j+\eta)}{\varphi(u_k-u_j-\eta)} \,, \qquad k=1,\dots,K \,. \tag{32}$$

**Shift operators**  It is also useful to define the shift operator

$$V = P_{12}P_{23}\cdots P_{2L-1,2L} \,, \tag{33}$$

which shifts all qubits towards the right by one site. We have

$$\hat{\Sigma}_k^{(N)}|\Omega\rangle = V^k \hat{\Sigma}_0^{(N)}|\Omega\rangle = V^k|\mathrm{DW}_N\rangle \,. \tag{34}$$

Due to our choice of inhomogeneities (24) which is alternating, the one-site shift operator $V$ does not commute with the evolution operator $\mathcal{U}$ defined in (1) and hence is not diagonalized by a generic Bethe state. In our context, it is more natural to define the two-site shift operator $\mathcal{V} = V^2$, which commutes with $\mathcal{U}$. Therefore, the Bethe states are also eigenstates of $\mathcal{V}$:

$$\mathcal{V}|\mathbf{u}_K\rangle = \mu(\mathbf{u}_K)|\mathbf{u}_K\rangle \,, \tag{35}$$

where the eigenvalue $\mu(\mathbf{u}_K)$ is given by

$$\mu(\mathbf{u}_K) = \prod_{k=1}^{K} \frac{\varphi(u_k-\frac{u}{2})}{\varphi(u_k-\frac{u}{2}+\eta)} \frac{\varphi(u_k+\frac{u}{2})}{\varphi(u_k+\frac{u}{2}+\eta)} \,. \tag{36}$$

Due to the periodic boundary condition, $\mu(\mathbf{u}_K)$ satisfies the following quantization condition

$$\mu(\mathbf{u}_K)^L = 1 \,. \tag{37}$$

**Overlaps** Inserting a resolution of the identity[3] in (16), we obtain

$$\mathcal{D}^{(N)}(n, 2k) = \sum_{\text{sol}_N} \frac{\tau(\mathbf{u}_N)^n}{\mu(\mathbf{u}_N)^k} \frac{\langle \mathrm{DW}_N | \mathbf{u}_N \rangle \langle \mathbf{u}_N | \mathrm{DW}_N \rangle}{\langle \mathbf{u}_N | \mathbf{u}_N \rangle}, \tag{38}$$

where 'sol$_N$' means that we take the sum over all physical solutions of the Bethe Ansatz equations (32) with $N$ Bethe roots. The reason that we only need to sum over the $N$-magnon sector is that $|\mathrm{DW}_N\rangle$ only has non-zero overlap with $N$-magnon states, due to $S^z$ conservation. To write the result in terms of the Bethe roots, we also need analytic expressions for the overlaps $\langle \mathrm{DW}_N | \mathbf{u}_N \rangle$, $\langle \mathbf{u}_N | \mathrm{DW}_N \rangle$ and the norm $\langle \mathbf{u}_N | \mathbf{u}_N \rangle$. Fortunately, these quantities have been computed [43–46]. We summarize the results below.

The overlap is given by

1. Even $N$

$$\langle \mathrm{DW}_N | \mathbf{u}_N \rangle = \prod_{k=1}^{N} \varphi(u_k - \tfrac{u}{2} + \eta)^{N/2} \varphi(u_k + \tfrac{u}{2} + \eta)^{N/2} \frac{\det H^{\text{even}}}{\varphi(u)^{N^2/4} \prod_{j>k} \varphi(u_j - u_k)}, \tag{39}$$

where $H^{\text{even}}$ is an $N \times N$ matrix whose matrix elements are given by

$$H_{ab}^{\text{even}} = \begin{cases} g_b(u_a, \tfrac{u}{2}), & 1 \le b \le \tfrac{N}{2}, \\ g_{b-\frac{N}{2}}(u_a, -\tfrac{u}{2}), & \tfrac{N}{2} < b \le N, \end{cases} \tag{40}$$

with

$$g_n(u, \theta) = \frac{\varphi'(u - \theta)}{\varphi(u - \theta)} - \frac{\varphi'(u - \theta + \eta)}{\varphi(u - \theta + \eta)}. \tag{41}$$

2. Odd $N$

$$\langle \mathrm{DW}_N | \mathbf{u}_N \rangle = \prod_{k=1}^{N} \varphi(u_k - \tfrac{u}{2} + \eta)^{\frac{N+1}{2}} \varphi(u_k + \tfrac{u}{2} + \eta)^{\frac{N-1}{2}} \frac{\det H^{\text{odd}}}{\varphi(u)^{(N^2-1)/4} \prod_{j>k}^{N} \varphi(u_j - u_k)}, \tag{42}$$

where $H^{\text{odd}}$ is an $N \times N$ matrix whose matrix elements are given by

$$H_{ab}^{\text{odd}} = \begin{cases} g_b(u_a, \tfrac{u}{2}), & 1 \le b \le \tfrac{N+1}{2}, \\ g_{b-\frac{N+1}{2}}(u_a, -\tfrac{u}{2}), & \tfrac{N+1}{2} < b \le N, \end{cases} \tag{43}$$

and the function $g_a(u, \theta)$ is given in (41).

The norm of the on-shell Bethe state is given by

$$\langle \mathbf{u}_N | \mathbf{u}_N \rangle = \varphi(\eta)^N \prod_{k=1}^{N} \left( \frac{\varphi(u_k - \tfrac{u}{2})}{\varphi(u_k - \tfrac{u}{2} + \eta)} \right)^{2L} \left( \frac{\varphi(u_k + \tfrac{u}{2})}{\varphi(u_k + \tfrac{u}{2} + \eta)} \right)^{2L} \prod_{j \ne k}^{N} \frac{\varphi(u_j - u_k + \eta)}{\varphi(u_j - u_k)} \times \det G, \tag{44}$$

where $G$ is the Gaudin matrix whose elements are

$$G_{ab} = -\frac{\partial}{\partial u_b} \ln \left[ \left( \frac{\varphi(u_a - \tfrac{u}{2} + \eta)}{\varphi(u_a - \tfrac{u}{2})} \right)^{L} \left( \frac{\varphi(u_a + \tfrac{u}{2} + \eta)}{\varphi(u_a + \tfrac{u}{2})} \right)^{L} \right] - \frac{\partial}{\partial u_b} \ln \left[ \prod_{k=1}^{N} \frac{\varphi(u_a - u_k - \eta)}{\varphi(u_a - u_k + \eta)} \right]. \tag{45}$$

Finally we also need the overlap $\langle \mathbf{u}_N | \mathrm{DW}_N \rangle$. This is the complex conjugate of $\langle \mathrm{DW}_N | \mathbf{u}_N \rangle$. However, it is not convenient to work with complex conjugate in the computational algebraic geometry method. For even $N$, we have the following relation

$$\langle \mathbf{u}_N | \mathrm{DW}_N \rangle = \prod_{k=1}^{N} \left( \frac{\varphi(u_k - \tfrac{u}{2})}{\varphi(u_k - \tfrac{u}{2} + \eta)} \right)^{N/2} \left( \frac{\varphi(u_k + \tfrac{u}{2})}{\varphi(u_k + \tfrac{u}{2} + \eta)} \right)^{N/2} \langle \mathrm{DW}_N | \mathbf{u}_N \rangle. \tag{46}$$

---

[3]Such insertion relies on completeness of the Bethe Ansatz which will be discussed in Section 3.2.

The proof is given in Appendix A. Define the following quantity

$$w(\mathbf{u}_N) = \frac{\langle \mathrm{DW}_n | \mathbf{u}_N \rangle \langle \mathbf{u}_N | \mathrm{DW}_N \rangle}{\langle \mathbf{u}_N | \mathbf{u}_N \rangle}, \tag{47}$$

where the analytic expressions for the three overlaps have been given in (39), (46) and (44). Therefore our task is to compute the following quantity

$$\mathcal{D}^{(N)}(n, 2k) = \sum_{\mathrm{sol}_N} \frac{\tau(\mathbf{u}_N)^n}{\mu(\mathbf{u}_N)^k} w(\mathbf{u}_N), \tag{48}$$

for arbitrary $n$. The Fourier transform of this quantity can be computed readily

$$\widehat{\mathcal{D}}^{(N)}(\omega, Q) = \sum_{k=1}^{L} e^{ikQ} \sum_{n=0}^{\infty} e^{-n(i\omega+\gamma)} \mathcal{D}^{(N)}(n, 2k) \tag{49}$$

$$= \sum_{\mathrm{sol}_N} \left( \sum_{k=1}^{L} \frac{e^{ikQ}}{\mu(\mathbf{u}_N)^k} \right) \frac{w(\mathbf{u}_N)}{1 - e^{-i\omega-\gamma} \tau(\mathbf{u}_N)}.$$

Since $\mu(\mathbf{u}_N)$ are quantized, we can write it as

$$\mu(\mathbf{u}_N) = e^{iQ_0}, \qquad Q_0 = \frac{\pi m}{L}, \qquad m = 0, 1, \ldots, L-1. \tag{50}$$

If we fix the total quasi-momentum $Q = Q_0$, (49) simplifies to

$$\widehat{\mathcal{D}}^{(N)}(\omega, Q_0) = L \sum_{\mathrm{sol}_{N,Q_0}} \frac{w(\mathbf{u}_N)}{1 - e^{-i\omega-\gamma} \tau(\mathbf{u}_N)}, \tag{51}$$

where now the summation is over the physical solutions of the Bethe Ansatz equations with total quasi-momentum $Q_0$. This can be achieved by adding the additional constraint

$$e^{iQ_0} = \prod_{k=1}^{N} \frac{\varphi(u_k - \frac{u}{2})}{\varphi(u_k - \frac{u}{2} + \eta)} \frac{\varphi(u_k + \frac{u}{2})}{\varphi(u_k + \frac{u}{2} + \eta)} \tag{52}$$

to the BAE. This will in general reduce the number of allowed solutions.

## 3.2 Computational algebraic geometry

In order to compute (48) and (51), one of the main difficulties is summing over solutions of Bethe Ansatz equations. In principle, this can only be done numerically by solving BAE and plugging in the solutions into (48) and (51). However, there are drawbacks to this approach. The BAE and the physical quantities depend on the parameters $\eta$ and $u$. In order to solve the BAE numerically, we need to fix these parameters to certain values. On the other hand, computations of small lattice size show that the final results are rational functions in these parameters.[4] In many situations, it is much more desirable to have analytic expressions so that one can perform various analyses. Therefore, we would like to obtain results (48) and (51) that are as analytical as possible, with explicit $u$ or $\eta$ (or both) dependence.

To this end, we apply the algebro-geometric method which allows us to compute the sum in (48) and (51) over solutions of BAE analytically. This approach is completely analytic and exact, which avoids solving the equations explicitly. The efficiency of the computation depends

---

[4]More precisely, for the isotropic case (see details below (7)) the result are rational functions in $u$. For anisotropic the results are rational functions of $e^\eta$ and $e^u$.

on its implementations. With proper implementation, we can obtain analytic results for $L \sim 10$ (corresponding to 20 qubits[5]) with any magnon numbers. Our approach nicely covers the range of medium system size with magnon number $N$ ranging in $1, \ldots, L$, where brute force computations become infeasible while thermodynamic approximation is not always applicable.

Completeness of $Q$-system is equivalent to completeness of BAE. The later was proven for a generic values of inhomogeneties in [20, 26, 47–50].

**Rational $Q$-system**  To perform the sum in (48) and (51) analytically, there is one more difficulty. It is well-known that among solutions of BAE there exist ones that do not correspond to eigenstates of the transfer matrix; such solutions are called non-physical solutions. They have to be discarded while performing the sum. Deciding which solutions are physical is a non-trivial task and is related to the completeness problem of the Bethe Ansatz. In order to sum over only the physical solutions, we use the rational $Q$-system. The rational $Q$-system is a non-trivial reformulation of the BAE, which is more efficient than solving BAE.[6] More importantly, it leads to only physical solutions. This was first observed in [21] and was proven and generalized in [22, 23, 28]. For XXX and XXZ spin chains, the rational $Q$-systems have been constructed in [21] and [23] respectively. Completeness of the rational $Q$-system (which is equivalent to the so-called Wronskian Bethe Equations) has been proven in [48], based on earlier works [51, 52].

We give a brief introduction to the rational $Q$-system in Appendix C. Here it is sufficient to mention that, in the rational $Q$-system, we are solving a set of algebraic equations for the coefficients of Baxter's $Q$-polynomials. For a state with $N$ Bethe roots, Baxter's $Q$-polynomial for the XXX spin chain (see details below (7)) are defined as

$$Q_{\mathrm{XXX}}(u) = \prod_{j=1}^{N}(u - u_j) = u^N + \sum_{k=0}^{N-1} c_k u^k. \tag{53}$$

The rational $Q$-system will give a set of algebraic equations for $\{c_k\}$. For the XXZ chain, Baxter's polynomial is defined by

$$\widetilde{Q}_{\mathrm{XXZ}}(u) = \prod_{j=1}^{N} \sinh(u - u_j), \tag{54}$$

in terms of additive variables $u$ and $u_k$. For the XXZ case, it is more convenient to use the multiplicative variables

$$x = e^{2u}, \qquad x_k = e^{2u_k}, \qquad q = e^{\eta}, \qquad \xi = e^{2\theta}. \tag{55}$$

The $Q$-function in (54) becomes

$$\widetilde{Q}_{\mathrm{XXZ}}(u) \propto Q_{\mathrm{XXZ}}(x) = \prod_{j=1}^{N}(x - x_j) = x^N + \sum_{k=0}^{N-1} c_k x^k. \tag{56}$$

In the framework of the rational $Q$-system, one first derives a set of equations for the coefficients $\{c_k\}$ in (53) or (56). These algebraic equations are called the zero remainder conditions. One then either solves these equations explicitly, or manipulates these equations analytically using computational algebraic geometric approaches.

---

[5]Note that the number of qubits in the brickwork model is $2L$.

[6]We compare the efficiency for obtaining *all* physical solutions using the two different formulations. In terms of finding *specific* solutions, such as the antiferromagnetic vacuum ground state and the first few excited states, it is more convenient to use the original Bethe Ansatz equation (or more precisely, the logarithmic of the BAE). For these solutions, the mode number exhibit clear pattern and the Bethe roots can be found numerically by iteration.

**Physical quantities in $\{c_k\}$**   Once the coefficients $\{c_k\}$ are known, we have all the $Q$-polynomials of the $Q$-system (see Appendix C for details). To find the Bethe roots, we simply need to find the zeros $Q$-polynomial $Q_{0,1}(u)$. However, since our aim is computing physical quantities analytically, we can also avoid this procedure. All physical quantities are symmetric functions of Bethe roots. More precisely, the quantities we are considering are *symmetric rational functions* of Bethe roots. At the same time, from (53) it is clear that $\{c_k\}$ are essentially the elementary symmetric polynomials of $\{u_k\}$, *i.e.*

$$c_{N-1} = -(u_1 + u_2 + \ldots + u_N)\,, \tag{57}$$
$$c_{N-2} = u_1 u_2 + u_1 u_3 + \ldots + u_{N-1} u_N\,,$$
$$\vdots$$
$$c_0 = (-1)^N u_1 u_2 \cdots u_N\,.$$

Therefore it is clear that all the physical quantities can also be written in terms of $\{c_k\}$.

In the algebraic geometrical computation, we will work with expressions in terms of $\{c_k\}$. Since the overlaps are originally given in terms of the Bethe roots $\{u_k\}$, it is necessary to perform a change of variables from $\{u_k\}$ to $\{c_k\}$. When the expressions are relatively simple with a few variables, it is straightforward to perform the calculation.[7] However, for more involved expressions, *e.g.* the explicit analytic formula of the Gaudin determinant, such a procedure can be quite tedious. At the moment we do not have an analytic formula for Gaudin-like determinants directly written in terms of $\{c_k\}$. Nevertheless, we can perform the change of variables with the help of some tricks using the Gröbner basis, which accelerates the computation significantly. We will discuss this procedure in more detail in Appendix D.

**Computational algebraic geometry**   Now we discuss the main procedure for the algebro-geometric (AG) computations. More details can be found in Appendix B and the mathematical reference is [53].

Suppose we want to compute the following sum

$$S = \sum_{\text{sol}} \frac{P(x_1, \ldots, x_n)}{Q(x_1, \ldots, x_n)}\,, \tag{58}$$

where $P(x_1, \ldots, x_n)$ and $Q(x_1, \ldots, x_n)$ are polynomials in $x_1, \ldots, x_n$. Here $\{x_k\}$ are solutions of a set of polynomial equations

$$f_k(x_1, \ldots, x_n) = 0\,, \qquad k = 1, \ldots, N\,. \tag{59}$$

We assume that the system (59) has finitely many solutions, which is the case for Bethe equations for $\{u_k\}$, or equivalently the zero remainder conditions for $\{c_k\}$. To perform the sum in (58), usually we first solve the equations (59) and find all possible solutions, and then plug the solutions into the summand in (58) and finally perform the sum. In the algebraic geometry (AG) computation, the strategy is different. In order to explain the main idea, let us introduce a few notions in algebraic geometry (or commutative algebra). The set of all polynomials in $n$ variables $\{x_1, \ldots, x_n\}$ with complex coefficients forms a polynomial ring, which we denote by $\mathbb{C}[x_1, \ldots, x_n]$. The polynomials in (59) which define the polynomial equations generate an ideal

$$\mathcal{I} = \langle f_1, f_2, \ldots, f_N \rangle\,. \tag{60}$$

In other words, the ideal $\mathcal{I}$ consists of all polynomials of the form $\sum_i a_i f_i$ where $a_i \in \mathbb{C}[x_1, \ldots, x_n]$ are also polynomials. One important fact of the ideal $\mathcal{I}$ is that the same

---

[7]For example, using the function `SymmetricReduction` of MATHEMATICA.

ideal can be generated by different sets of polynomials. These polynomials are called bases of the ideal. The most important kind of basis for us is the so-called Gröbner basis (see details in Appendix B.2), which we will denote by $g_j$ and we have

$$\mathcal{I} = \langle f_1, f_2, \ldots, f_N \rangle = \langle g_1, g_2, \ldots, g_{N'} \rangle, \tag{61}$$

where in general $N' \neq N$. The Gröbner basis has the nice feature that polynomial reductions with respect to it have well-defined and unique remainders.

The quotient space

$$V_f = \mathbb{C}[x_1, \ldots, x_n]/\mathcal{I} \tag{62}$$

is called the quotient ring. The quotient ring is a finite dimensional vector space, and by a fundamental theorem in algebraic geometry [53], the dimension $\dim V_f = N_s$ equals to the number of solutions of equations (59). Since $V_f$ is an $N_s$-dimensional vector space, we can choose a basis which we denote by $e_1, \ldots, e_{N_s}$. There is a canonical choice of such a basis, which can be constructed from the Gröbner basis of $\mathcal{I}$. After introducing the canonical basis $\{e_j\}$, any polynomial $P \in \mathbb{C}[x_1, \ldots, x_n]$ can be mapped to a finite dimensional matrix $\mathbf{P}$ in $V_f$ by

$$P e_i \sim \sum_{j=1}^{N_s} [\mathbf{P}]_{ij} e_j, \tag{63}$$

where '$\sim$' means performing polynomial reduction with respect to the Gröbner basis of the ideal $\mathcal{I}$ and finding the remainder. The remainder can be written as a linear combination of the basis $\{e_j\}$ with coefficients that are matrix elements $[\mathbf{P}]_{ij}$. The $N_s \times N_s$ dimensional matrix $\mathbf{P}$ is called the *companion matrix* of the polynomial $P$ [53]. It is a linear representation of the polynomial algebra. Note that the companion matrix $\mathbf{P}$ no longer depends on the variables $\{x_1, \ldots, x_n\}$, but it can depend on other parameters such as the spectral parameter $u$ and the anisotropic parameter $\eta$ (for XXZ case) as discussed below.

To summarize, given the system of equations (59) and a polynomial $P(x_1, \ldots, x_n)$, we construct its companion matrix $\mathbf{P}$ by the following steps

1. Compute the Gröbner basis of the equations (59).

2. Using the Gröbner basis, construct the canonical basis of the quotient ring.

3. Using both the Gröbner basis and quotient ring basis, construct the companion matrix.

All the above computations can be automatized. For the computation of the Gröbner basis, we use the package `Singular`. The quotient ring and companion matrices can be computed using our own MATHEMATICA package BeF2, which is currently private. In certain computations for larger quantum numbers $L$ and $M$, similar procedures are also implemented with `Singular` and GPI-SPACE framework to boost the efficiency of certain computations of the companion matrix both over real and Fourier space.

Finally, in order to compute the sum (58), we construct companion matrices for $P$ and $Q$

$$P(x_1, \ldots, x_n) \mapsto \mathbf{P}, \qquad Q(x_1, \ldots, x_n) \mapsto \mathbf{Q}. \tag{64}$$

From computational algebraic geometry [53], the sum (58) can be computed analytically by

$$S = \mathrm{tr}\left[\mathbf{P} \cdot \mathbf{Q}^{-1}\right], \tag{65}$$

where $\mathbf{Q}^{-1}$ is the inverse matrix of $\mathbf{Q}$. Note that the computation via algebraic geometry does not need to solve the equation numerically. The result (65) is exact since no float number is used in the Gröbner basis or the companion matrices computations.

**A comprehensive example**   To demonstrate the application of the above theoretical discussion, a comprehensive example is provided here. Define an algebraic equation in variables $x, y, z$, as a toy model of Q-system equation,

$$f_1 = -x^3 - x + y^2 - 1, \qquad f_2 = x^2 + yz - 1, \qquad f_3 = -x^2 + xz + 3. \tag{66}$$

Let the ideal $I = \langle f_1, f_2, f_3 \rangle$. The solution set of $f_1(x, y, z) = f_2(x, y, z) = f_3(x, y, z) = 0$, or zero locus of $I$ in the language of algebraic geometry, is denoted as $\mathcal{S}$. As a simulation of a physical computation in the Q-system, in this toy model, the goal is to calculate the sum over roots in $\mathcal{S}$,

$$\sum_{p \in \mathcal{S}} \frac{P(p)}{Q(p)}, \tag{67}$$

where $P$ and $Q$ are polynomials. Take $P = x^2 - y^2$ and $Q = x^2 + y^2 + z^2$ as an example. With the software SINGULAR [54], the Gröbner basis of the ideal is obtained,

$$\begin{aligned}
G(I) = \{ & xz + yz + 2, 2xy - 12x + 3y^2 - z^2 + 3z - 5, x^2 + yz - 1, \\
& 2x - y^2 - 12yz + 6y + z^3 - 3z^2 + 5z - 23, -4x + y^2 + yz^2 - z - 1, \\
& 2x + y^2z - y^2 + 2y + 1, -26x + y^3 + 7y^2 - yz - 3y - 2z^2 + 5z - 7 \}.
\end{aligned} \tag{68}$$

From the lead monomials of $G(I)$, $\mathbb{C}[x, y, z]/I = \mathrm{span}_{\mathbb{C}}\{y^2, yz, z^2, x, y, z, 1\}$ and the solution set must consist of 7 points from algebraic geometry [53]. Furthermore, the companion matrices of $F$ and $G$ can be calculated from (63),

$$M_P = \begin{pmatrix}
-5 & 5 & 0 & 0 & 6 & 0 & -1 \\
11 & 0 & 4 & 3 & -1 & 0 & -1 \\
0 & -1 & 0 & 0 & -2 & 0 & 0 \\
6 & -12 & 0 & 0 & -24 & -2 & 0 \\
14 & 0 & 0 & 2 & 0 & 2 & 0 \\
4 & 3 & 0 & 1 & 5 & 0 & 0 \\
-19 & -5 & 4 & -6 & -6 & 0 & 1
\end{pmatrix}, \tag{69}$$

and

$$M_Q = \begin{pmatrix}
20 & -8 & -6 & 3 & -9 & 3 & 1 \\
-13 & -2 & 36 & -3 & 1 & 12 & -1 \\
-2 & 2 & 4 & 0 & 2 & 3 & 1 \\
-36 & 18 & 36 & -6 & 32 & -8 & 0 \\
-20 & 6 & -24 & -2 & 6 & -8 & 0 \\
-1 & 0 & 18 & -4 & -4 & -5 & 0 \\
18 & -8 & 94 & 3 & 9 & 21 & 1
\end{pmatrix}. \tag{70}$$

By the trace formula (65),

$$\sum_{p \in \mathcal{S}} \frac{P(p)}{Q(p)} = \mathrm{tr}(M_P M_Q^{-1}) = -\frac{357867}{63883}. \tag{71}$$

This is the exact result from computational algebraic geometry. Note that here we *never* solve the equation to find the roots. As a comparison, a numeric computation can find the approximate values of the 7 solutions, and the sum (67) has the value $-5.60191\ldots$ which approximates the exact result. So the computation algebraic geometry method gets the exact sum, without solving the equation or introducing algebraic extensions.

**Dealing with free parameters**    Before ending the section, we make a remark. One of the main challenges in the current context for the AG computation is that our equations have free parameters $u$ and $\eta$. In principle, the computational algebraic geometry algorithm also works for such cases. In practice, however, the procedure usually involves bulky intermediate expressions which slow down the computation. Therefore, we need a better way to implement the computation. The strategy we take is to perform the computation for different rational values of the parameters and then fit the analytic final result. As will be shown below, the correlation functions are rational functions of $u$ in the XXX case and are rational functions of $e^u$ and $e^\eta$ in the XXZ case. In both cases, we can factorize out simple common denominators, after which we only need to fit polynomials.

## 4  Exact results I. Real space

In this section, we present analytic results for the correlation function

$$\mathcal{D}^{(N)}(n,k) = \langle\Omega|\hat{\Sigma}_k^{(N)}\mathcal{U}^n\hat{\Sigma}_0^{(N)}|\Omega\rangle\,. \tag{72}$$

We present the results for $k = 0$, in which case $\mathcal{D}^{(N)}(n,0)$ is essentially a discrete version of the Loschmidt echo. Results for $k \neq 0$ can be obtained easily because the shift in $k$ only leads to a global phase factor $e^{ikQ_0}$. Written in terms of rapidities, it is given by

$$\mathcal{D}^{(N)}(n,0) = \sum_{\text{sol}_N} \tau(\mathbf{u}_N)^n\, w(\mathbf{u}_N)\,. \tag{73}$$

Before presenting the result, let us discuss the general structure. From the definition (72), it is clear that the final result is given by certain matrix elements of $\mathcal{U}^n$. From the definition of the discrete two-step time evolution operator $\mathcal{U}$ (23), matrix elements are polynomials of $b(u)$ and $c(u)$ where $b(u) = \varphi(u)/\varphi(u+\eta)$ and $c(u) = \varphi(\eta)/\varphi(u+\eta)$ are given in (7). The result depends on three global quantum numbers: the size of the circuit, which we denote by $2L$; the length of the string of operators $\hat{\Sigma}^{(N)}$ which is denoted by $N$; and the evolution time denoted by $n$. Therefore we find the results take the following form

$$\mathcal{D}_L^{(N)}(n) = \sum_{l,k}\alpha_{l,k}\left(\frac{\varphi(u)}{\varphi(u+\eta)}\right)^l\left(\frac{\varphi(\eta)}{\varphi(u+\eta)}\right)^k\,, \qquad l,k \geq 0\,, \tag{74}$$

where $\alpha_{l,k}$ are non-negative integers. In the rational limit, $\varphi(u) \to u$ and $\eta \to i$, we have, correspondingly,

$$\mathcal{D}_L^{(N)}(n) = \sum_{l,k}\alpha_{l,k}\left(\frac{u}{u+i}\right)^l\left(\frac{i}{u+i}\right)^k\,, \qquad l,k \geq 0\,. \tag{75}$$

Therefore, in the isotropic limit, the result is a rational function of $u$. The AG computation leads to such rational functions. Below we will discuss the results in the isotropic case (75). Once we obtain the results in the rational limit using the AG computation, we can solve for the coefficients $\alpha_{l,k}$. Plugging $\alpha_{l,k}$ back into (74) gives the result for generic $\eta$.

### 4.1  Results for small quantum numbers

For small values of $L, N, n$, we can compute $\mathcal{D}_L^{(N)}(n)$ by brute force, without using Bethe Ansatz. These results can serve as benchmarks for the Bethe Ansatz and AG computation. They already exhibit some interesting features. We list the results for $2L = 4, 6, 8$ below for small $N$ and $n$.

**Results for $L = 2$.** For $N = 1$, we have

$$\mathcal{D}_2^{(1)}(1) = c^2,$$ (76)
$$\mathcal{D}_2^{(1)}(2) = (b^2 + c^2)^2,$$
$$\mathcal{D}_2^{(1)}(3) = (3b^2c + c^3)^2,$$
$$\mathcal{D}_2^{(1)}(4) = (b^4 + 6b^2c^2 + c^4)^2,$$
$$\mathcal{D}_2^{(1)}(5) = (5b^4c + 10b^2c^3 + c^5)^2,$$

where $b, c$ are short for $b(u)$ and $c(u)$. For $N = 2$, we obtain

$$\mathcal{D}_2^{(2)}(1) = c^2,$$ (77)
$$\mathcal{D}_2^{(2)}(2) = 2b^4c^2 + b^4 + 2b^2c^4 + c^4,$$
$$\mathcal{D}_2^{(2)}(3) = 2b^8c^2 + 6b^6c^4 + 4b^6c^2 + 6b^4c^6 + 16b^4c^4 + 3b^4c^2 + 2b^2c^8 + 4b^2c^6 + c^6,$$
$$\mathcal{D}_2^{(2)}(4) = b^8 + 6b^8c^2 + 4b^{10}c^2 + 2b^{12}c^2 + 6b^4c^4 + 42b^6c^4 + 40b^8c^4 + 10b^{10}c^4 + 42b^4c^6$$
$$+ 72b^6c^6 + 20b^8c^6 + c^8 + 6b^2c^8 + 40b^4c^8 + 20b^6c^8 + 4b^2c^{10} + 10b^4c^{10} + 2b^2c^{12}.$$

Note that the results are not necessarily homogeneous polynomial in $b, c$.

**Results for $L = 3$.** For $N = 1$, we have

$$\mathcal{D}_3^{(1)}(1) = c^2,$$ (78)
$$\mathcal{D}_3^{(1)}(2) = 2b^2c^2 + c^4,$$
$$\mathcal{D}_3^{(1)}(3) = b^6 + 3b^4c^2 + 6b^2c^4 + c^6,$$
$$\mathcal{D}_3^{(1)}(4) = 12b^6c^2 + 18b^4c^4 + 12b^2c^6 + c^8.$$

For $N = 2$, we have

$$\mathcal{D}_3^{(2)}(1) = c^2,$$ (79)
$$\mathcal{D}_3^{(2)}(2) = b^4c^2 + 2b^2c^4 + c^4,$$
$$\mathcal{D}_3^{(2)}(3) = b^8 + 4b^8c^2 + 12b^6c^4 + 4b^4c^4 + 9b^4c^6 + 4b^2c^6 + 2b^2c^8 + c^6.$$

**Results for $L = 4$** For $N = 1$, we have

$$\mathcal{D}_4^{(1)}(1) = c^2,$$ (80)
$$\mathcal{D}_4^{(1)}(2) = 2b^2c^2 + c^4,$$
$$\mathcal{D}_4^{(1)}(3) = 3b^4c^2 + 6b^2c^4 + c^6.$$

For $N = 2$, we have

$$\mathcal{D}_4^{(2)}(1) = c^2,$$ (81)
$$\mathcal{D}_4^{(2)}(2) = b^4c^2 + 2b^2c^4 + c^4.$$

## 4.2 Short time behavior

At short times, the system does not evolve much and cannot 'feel' the size of the system. Therefore we expect that for small $n$, the result should be independent of the length $L$. This is

indeed what we observed from our analytic results. Let us illustrate this using $N = 2$ examples. Larger magnon numbers exhibit similar behaviors. For $N = 2$, we find that

$$\mathcal{D}_2^{(2)}(1) = \mathcal{D}_3^{(2)}(1) = \ldots = \mathcal{D}_L^{(2)}(1) = -\frac{1}{(u+i)^2}, \qquad L \geq 2. \tag{82}$$

Namely the result is the same for $n = 1$ for all $L \geq 2$. Similarly, we find for $n = 2$

$$\mathcal{D}_3^{(2)}(2) = \mathcal{D}_4^{(2)}(2) = \ldots = \mathcal{D}_L^{(2)}(2) = \frac{-u^4 + 3u^2 + 2iu - 1}{(u+i)^6}, \qquad L \geq 3, \tag{83}$$

and

$$\begin{aligned} \mathcal{D}_4^{(2)}(3) = \mathcal{D}_5^{(2)}(3) = \ldots &= \mathcal{D}_L^{(2)}(3), \qquad L \geq 4 \\ &= \frac{-u^8 + 12u^6 + 8iu^5 - 18u^4 - 12iu^3 + 12u^2 + 4iu - 1}{(u+i)^{10}}. \end{aligned} \tag{84}$$

In general, the pattern is that for any $n \geq 1$, the quantity $\mathcal{D}_L^{(2)}(n)$ is independent of $L$ for $L \geq n$. In other words, this indicates that in $n$ discrete steps, the system evolves at most to size $2n$. If the system is large enough, this spread does not touch the boundary of the system and therefore the result is independent of $L$, even though the intermediate calculations depends on $L$. This observation is consistent with the general fact that under short-range interactions, the propagation of information is limited in a light-cone structure governed by the Lieb-Robinson bound.

## 4.3 Long time behavior

As one can envisage, the quantity $\mathcal{D}_L^{(N)}(n)$ will be more involved as the discrete time $n$ grows. For large $n$, the analytic result we obtain is a rather complicated rational function of $u$. The result can be computed by the AG method, which is much more efficient than a brute force calculation. In this Subsection and in later parts of this work, we present results obtained by the AG methods.

In order to gain some idea what the result looks like, we present an example with $L = 8$, $N = 2$ and $n = 10$

$$\mathcal{D}_8^{(2)}(10) = \frac{P(u)}{(u+i)^{38}}, \tag{85}$$

where $P(u)$ is the following polynomial

$$\begin{aligned} P(u) = &-1 + 18iu + 243u^2 - 1776iu^3 - 12345u^4 + 57576iu^5 + 276288u^6 \\ &- 939576iu^7 - 3394500u^8 + 8772432iu^9 + 25101788u^{10} \\ &- 50778808iu^{11} - 119366908u^{12} + 196227896iu^{13} + 382479184u^{14} \\ &- 523609032iu^{15} - 837663006u^{16} + 961297284iu^{17} + 1247817562u^{18} \\ &- 1194362808iu^{19} - 1250264310u^{20} + 986339608iu^{21} + 831213408u^{22} \\ &- 531759240iu^{23} - 361426868u^{24} + 183920576iu^{25} + 101357548u^{26} \\ &- 40023464iu^{27} - 18065212u^{28} + 5290568iu^{29} + 1977488u^{30} \\ &- 391416iu^{31} - 121065u^{32} + 13530iu^{33} + 3435u^{34} - 120iu^{35} - 25u^{36}. \end{aligned} \tag{86}$$

Although $n = 10$ is not a very large number, the polynomial $P(u)$ is already rather bulky. For larger values of $n$, say around $n = 100$, the result involves huge polynomials in $u$. Presenting the results explicitly is not very illuminating. In order to gain more intuitions about the long

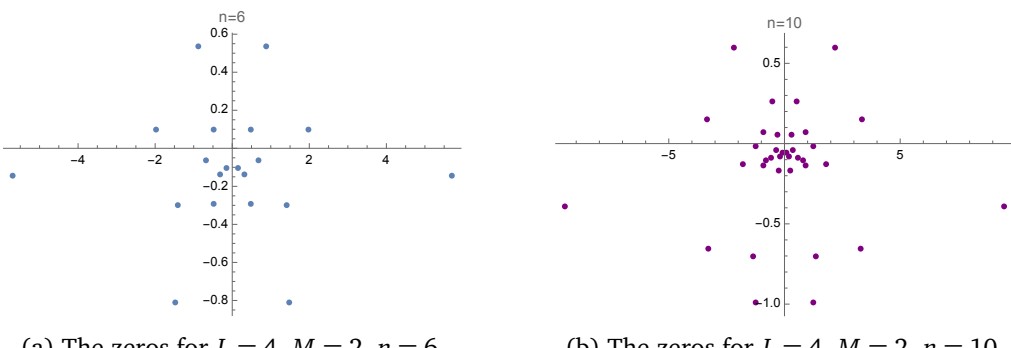

(a) The zeros for $L = 4$, $M = 2$, $n = 6$.      (b) The zeros for $L = 4$, $M = 2$, $n = 10$.

Figure 2: The zeros of $\mathcal{D}_4^{(2)}(6)$ and $\mathcal{D}_4^{(2)}(10)$.

time behavior of the Loschmidt echo, we instead plot the zeros of the corresponding $\mathcal{D}_L^{(N)}(n)$ as a function of $u$. These zeros are akin to the Lee–Yang zeros of thermal partition functions. It is known that the Lee–Yang zeros encode interesting physics of the model such as the phase structure. Here we also want to investigate how the zeros of $\mathcal{D}_L^{(N)}(n)$ change with $n$.

In Figure 2a and 2b, we plot the zeros of $\mathcal{D}_4^{(2)}$ for $n = 6$ and $n = 10$. We can see that the zeros both distribute symmetrically about the imaginary axis. The two distributions of zeros look somewhat similar but do not exhibit a clear pattern. However, as $n$ increases, the zeros will start to condense and a clear pattern emerges, as is shown in Figure 3. This phenomenon, namely the condensation of zeros of the $\mathcal{D}_L^{(M)}(n)$ for large $n$ is also manifested for other quantum numbers, with different condensation curves. In Figure 4, we show the zeros of $\mathcal{D}_4^{(4)}(500)$. Comparing the condensation curves of $\mathcal{D}_4^{(2)}(500)$ and $\mathcal{D}_4^{(4)}(500)$, we find that the condensation curves have a more complicated structure for larger $M$. These 'condensation

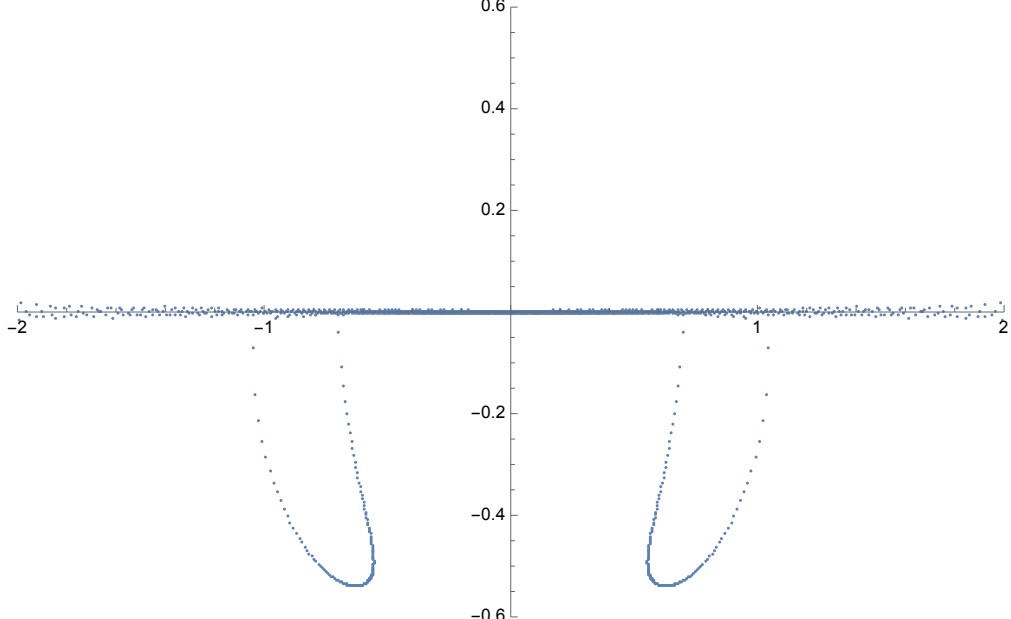

Figure 3: The zeros of $\mathcal{D}_4^{(2)}(500)$. There are 1998 zeros in total and most of them condense on three curves as can be seen clearly, one along the real axis and the other two in the lower half plane.

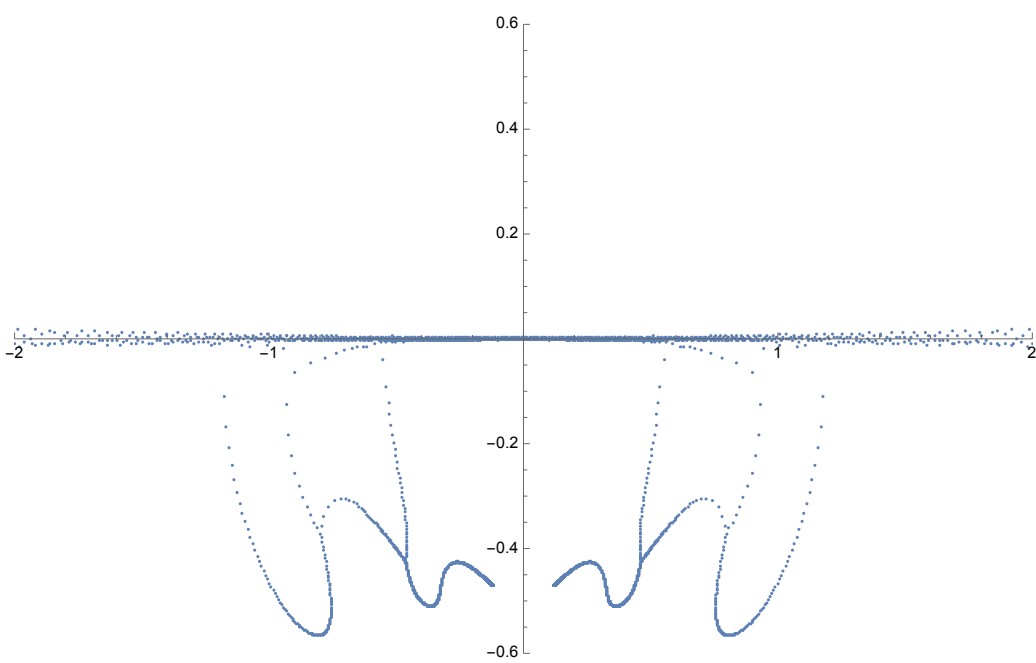

Figure 4: The zeros of $\mathcal{D}_4^{(4)}(500)$. There are 3992 zeros in total and they clearly condense on a few condensation curves, one along the real axis and two others in the lower half plane.

curves' can be defined more rigorously as follows [55]. We define the zero set of $\mathcal{D}_L^{(M)}(n)$ by

$$\mathcal{Z}\left(\mathcal{D}_L^{(M)}(n)\right) = \{u : \mathcal{D}_L^{(M)}(n) = 0\}.$$

In the limit $n \to \infty$, we can define the following limiting zero sets

$$\lim_{n\to\infty} \inf \mathcal{Z}\left(\mathcal{D}_L^{(M)}(n)\right) = \big\{u : \text{every neighborhood } u \in U \text{ has a nonempty}$$
$$\text{intersection with all but finitely many of the sets } \mathcal{D}_L^{(M)}(n)\big\},$$
$$\lim_{n\to\infty} \sup \mathcal{Z}\left(\mathcal{D}_L^{(M)}(n)\right) = \big\{u : \text{every neighborhood } u \in U \text{ has a nonempty}$$
$$\text{intersection with infinitely many of the sets } \mathcal{D}_L^{(M)}(n)\big\}.$$

According to the Beraha–Kahane–Weiss theorem which will be discussed below, in the limit $n \to \infty$,

$$\lim_{n\to\infty} \inf \mathcal{Z}\left(\mathcal{D}_L^{(M)}(n)\right) = \lim_{n\to\infty} \sup \mathcal{Z}\left(\mathcal{D}_L^{(M)}(n)\right),$$

and the two limiting zero sets actually coincides. This limiting set in our case takes the form of curves which are the condensation curves. For finite but large enough $n$, most of the zeros accumulate around the limiting condensation curve. Finally we comment that finding zeros of a high degree polynomial is a challenging numerical problem. We used the software MPSolve [56] which is a multiprecision implementation of the Aberth method [57].

## 4.4 A different initial state

The distribution of zeros that we have seen in the previous subsection is very similar to the zeros of the partition function of the lightcone 6-vertex model. In fact, we can identify the two

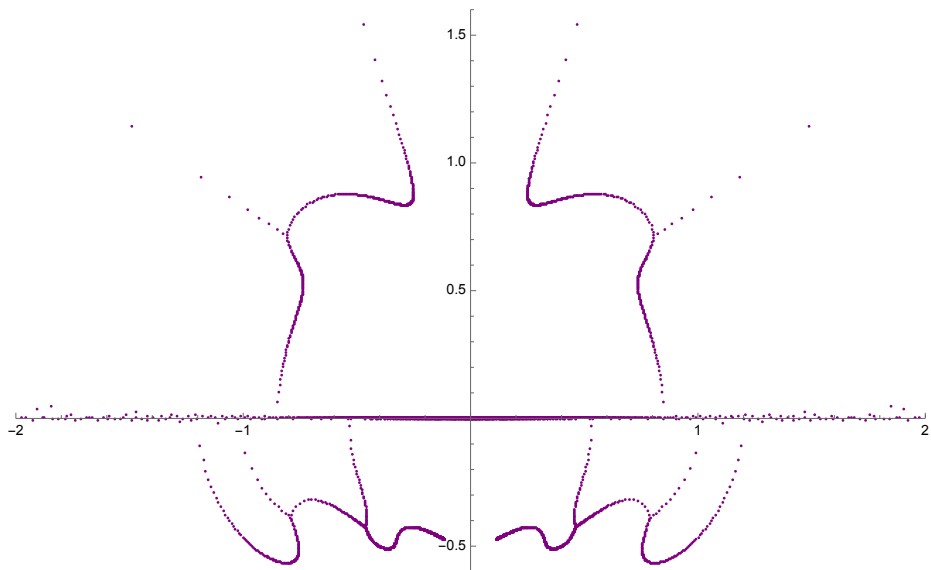

Figure 5: The zeros of $\widetilde{\mathcal{D}}_4(500)$, there are 3996 zeros in total, the zeros also condensates on a few condensation curves. The condensation curve is similar to the one of $\mathcal{D}_4^{(4)}(500)$. An obvious difference is that it also has condensation curves on the upper half plane.

quantities as follows. Let us consider the following state

$$\widetilde{\Sigma}|\Omega\rangle = (|01\rangle - |10\rangle)^{\otimes L} = \prod_{j=1}^{L}\left(\sigma_{2j-1}^x - \sigma_{2j}^x\right)|00\cdots 0\rangle\,, \tag{87}$$

which is the so-called dimer state. We can define a similar correlation function to (72)

$$\widetilde{\mathcal{D}}_L(n) = \langle\Omega|\widetilde{\Sigma}\,\mathcal{U}^n\,\widetilde{\Sigma}|\Omega\rangle\,. \tag{88}$$

Interestingly, the above correlation function $\widetilde{\mathcal{D}}_L(n)$ coincides with the cylinder partition function (with fixed boundaries at the two ends) of the 6-vertex model (up to a simple normalization factor) in statistical mechanics. For the rational case, it has been studied in [32] and the zeros of the partition functions have been studied in the partial thermodynamic limit. We have the following identification $\widetilde{\mathcal{D}}_L(n) \propto Z_{L,n}$ where $Z_{L,n}(u)$ is the partition function with size $L$ and $n$ in two directions. $Z_{L,n}(u)$ is a polynomial in $u$ for properly chosen normalizations. The partial thermodynamic limit corresponds to the limit with fixed finite $L$ and large $n$, which corresponds precisely to the long time behavior of the correlation function (88). One finds that the zeros of $Z_{L,n}(u)$ or equivalently $\widetilde{\mathcal{D}}_L(n)$ also condense on certain curves for large $n$. We compute the zeros of $\widetilde{\mathcal{D}}_L(n)$ for $L = 4$, $n = 500$ in Figure 5. In statistical mechanics, the condensation curves separate different phases of the model. One can even plot the condensation curves using numerical approaches and find a nice agreement with the distribution of the zeros [32].

Interestingly, we find that the condensation curves of $\widetilde{\mathcal{D}}_4(500)$ and $\mathcal{D}_4^{(4)}(500)$ are very similar in the lower half plane. This can be seen clearly by putting the two Figures on top of each other, as is shown in Figure 6. However, there are no condensation curves on the upper half plane for $\mathcal{D}_4^{(4)}(500)$.[8] The similarity between the condensation curves on the lower

---

[8]Note that this does not mean that there are no roots with positive imaginary parts. It means that the majority of such roots are distributed very near the real axis. For example, there are only 50 roots of $\mathcal{D}_4^{(4)}(500)$ with imaginary part larger 0.15, which constitute around 1% of the roots. As a comparison, there are 1704 roots with imaginary parts which are smaller than $-0.15$, which constitute 42% of the roots.

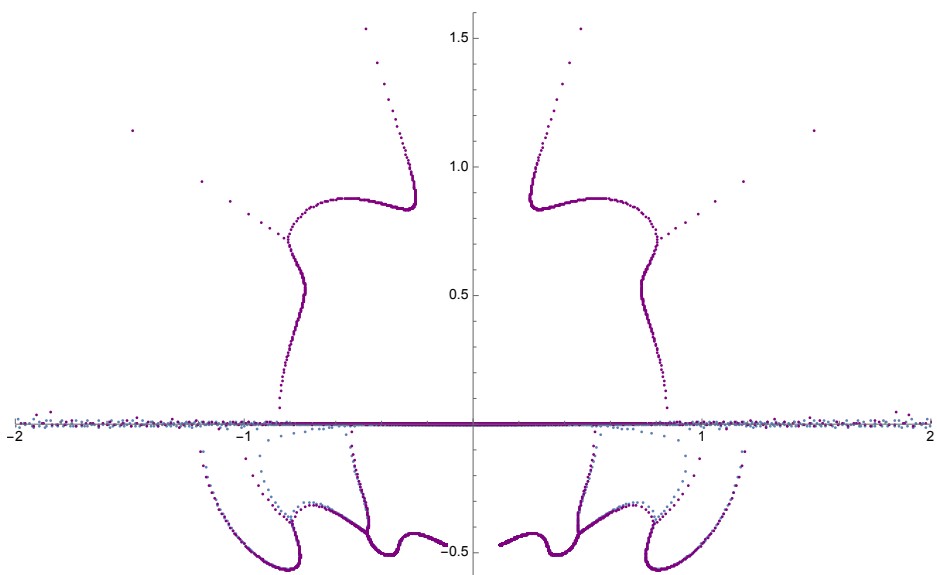

Figure 6: Comparison of the zeros of $\widetilde{\mathcal{D}}_4(500)$ and $\mathcal{D}_4^{(4)}(500)$. The purple and blue dots correspond to the zeros of $\widetilde{\mathcal{D}}_4(500)$ and $\mathcal{D}_4^{(4)}(500)$ respectively. We see that the condensation curves in the lower half plane almost coincide with each other, except that the zeros of $\mathcal{D}_4^{(4)}$ seem to have two small extra branches around $\pm[0.5, 0.9]$ close to the real axis.

half plane can be understood as follows. The appearance of condensation curve in the long time limit is a consequence of the Beraha–Kahane–Weiss theorem [55], which applies to the expressions of the form

$$\mathcal{Z}_n(u) = \sum_i \alpha(u)\Lambda_i(u)^n, \tag{89}$$

where we shall refer to $\Lambda_i(u)$ as the eigenvalues and $\alpha_i(u)$ as the corresponding multiplicities. Comparing to (73), we see that in the current context, $\Lambda_i(u) = \tau(\mathbf{u}_N)$ are the eigenvalues of the evolution operator $\mathcal{U}$ and the corresponding multiplicity $\alpha_i(u) = w(\mathbf{u}_N)$. We can arrange the eigenvalues according to the norm as follows

$$|\Lambda_1(u)| \geq |\Lambda_2(u)| \geq \dots \tag{90}$$

We say an eigenvalue is dominant if its norm is maximal. The BKW theorem states that in the $n \to \infty$ limit, the zeros of $\mathcal{Z}_n(u)$ will either form isolated points or curves. An isolated accumulation point occurs at $u = z_0$ if there is a unique dominant eigenvalue, namely $|\Lambda_1(z_0)| > |\Lambda_2(z_0)|$. A curve of accumulation points occurs when there are at least two dominant eigenvalues $|\Lambda_2(u)| = |\Lambda_2(u)|$ and the relative phase $\phi(u) \in \mathbb{R}$ defined by $\Lambda_2(u) = e^{i\phi(u)}\Lambda_1(u)$ varies along the curve. Therefore, the condensation curve will appear where there are at least two eigenvalues dominant. Both $\widetilde{\mathcal{D}}_4(n)$ and $\mathcal{D}_4^{(4)}(n)$ are computed in the $L = 8, M = 4$ sector and share the same set of eigenvalues $\Lambda_i(u)$. This accounts for the partial overlap observed in the condensation curves. However, there are also significant differences between the two cases. The dimer state (87) represents a special class of states known as *integrable boundary states*, whereas the domain wall state does not. Integrable boundary states possess higher symmetries compared to generic states in the Hilbert space and are annihilated by all odd higher charges of the model (see [58] for a more detailed discussion). Consequently, the overlap between an integrable boundary state and the eigenstates of the transfer matrix adheres to a specific selection rule: the overlap is non-zero only when the Bethe roots are parity

invariant, *i.e.*, $\{\mathbf{u}_N\} = \{-\mathbf{u}_N\}$. As a result, in the $\widetilde{\mathcal{D}}_4(n)$ case, only eigenvalues $\Lambda_i(u)$ associated with parity-invariant Bethe roots contribute, whereas in the $\mathcal{D}_4(n)$ case, all eigenvalues $\Lambda_i(u)$ contribute non-trivially. This distinction elucidates the differences in the condensation curves between the two cases.

An important insight from this analysis is that the condensation curves of the Lee-Yang zeros vividly reflect the symmetry of the initial (or final) states. Given that these curves demarcate different phases of the model, it follows that the symmetry of the initial states can influence the model's phase diagram. This observation opens an intriguing avenue for further investigation, which we leave for future work.

## 5 Exact results II. Fourier space

In this section, we consider the result of exact correlation functions in Fourier space

$$\widehat{\mathcal{D}}^{(N)}(\omega, Q_0) = L \sum_{\text{sol}_{N,Q_0}} \frac{w(\mathbf{u}_N)}{1 - e^{-i\omega - \gamma} \tau(\mathbf{u}_N)} \,. \tag{91}$$

The sum is now performed over the $N$ magnon sector with definite quasi-momentum $Q_0$. For concreteness, we take the value $Q_0 = 0$, which is the zero momentum sector for the given magnon number $N$. Other momentum sectors can be computed similarly. The AG computation procedure is similar, but the dimension of the quotient ring is smaller than the case without constraints on $Q_0$ because we focus on the zero momentum sector. Let us denote the companion matrices of $w(\mathbf{u}_N)$ and $\tau(\mathbf{u}_N)$ by

$$w(\mathbf{u}_N) \mapsto \mathbf{M}_w, \qquad \tau(\mathbf{u}_N) \mapsto \mathbf{M}_\tau \,. \tag{92}$$

The analytic result is given by

$$\widehat{\mathcal{D}}_L^{(N)}(\omega, 0) = L \operatorname{tr} \left[ \mathbf{M}_w \cdot (1 - a \mathbf{M}_\tau)^{-1} \right], \qquad a = e^{-i\omega - \gamma} \,. \tag{93}$$

The quantity $\widehat{\mathcal{D}}_L^{(N)}(\omega, 0)$ is the one studied in [18] numerically. We consider the generic XXZ $\check{R}$-matrix. In general, the result depends on $\xi = e^u$, $q = e^\eta$ and $a = e^{-i\omega - \gamma}$ and is a rational function of these three variables. Even for relatively small quantum numbers, the explicit analytic result is rather involved. As an example, the analytic result for $L = 4$, $M = 2$ reads

$$\widehat{\mathcal{D}}_2^{(2)}(\omega, 0) = \frac{N_{2,2}(\xi, q, a)}{D_{2,2}(\xi, q, a)} \,, \tag{94}$$

where

$$\begin{aligned} N_{2,2}(\xi, q, a) = {} & -a^2(q - \xi)^2(\xi + q)^2\left(\xi^2 + q^2\left(\xi^4 + \xi^2\left(q^2 - 4\right) + 1\right)\right) \\ & + a\left(\xi^2 + q^2\left(\xi^4 + \xi^2\left(q^2 - 4\right) + 1\right)\right)\left(\xi^2 + q^2\left(\xi^4\left(q^2 + 1\right) + \xi^2\left(q^2 - 6\right) + 1\right) + 1\right) \\ & - \left(\xi^2 q^2 - 1\right)^4 \,, \end{aligned} \tag{95}$$

and

$$\begin{aligned} D_{2,2}(\xi, q, a) = {} & a^3\left(q^2 - \xi^2\right)^4 \\ & - a^2\left(2\xi^4 + \xi^2 + \left(\xi^2 + 2\right)q^4 + \left(\xi^4 - 8\xi^2 + 1\right)q^2\right)\left(\xi^2 + q^2\left(\xi^4 + \xi^2\left(q^2 - 4\right) + 1\right)\right) \\ & + a\left(\xi^2 + q^2\left(\xi^4 + \xi^2\left(q^2 - 4\right) + 1\right)\right)\left(\xi^2 + q^2\left(\xi^4\left(2q^2 + 1\right) + \xi^2\left(q^2 - 8\right) + 1\right) + 2\right) \\ & - \left(\xi^2 q^2 - 1\right)^4 \,. \end{aligned} \tag{96}$$

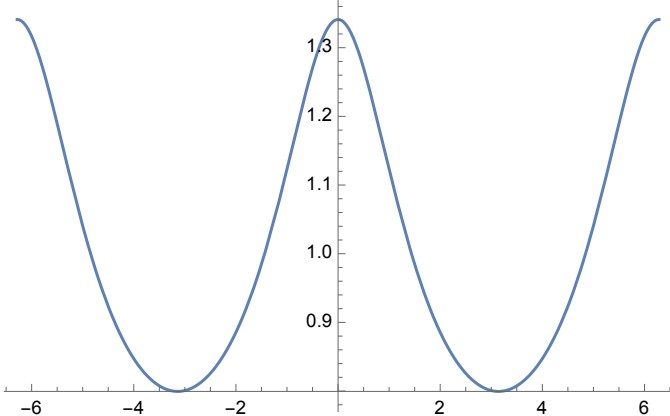

Figure 7: Plot of $|\widehat{\mathcal{D}}_2^{(2)}(\omega, 0)|$ with parameters $\xi = 9$ and $q = 2$. We plot in the range $-2\pi \leq \omega \leq 2\pi$.

In what follows, we will fix $\xi$ and $q$ to certain values and present the result as a rational function of $a$. We distinguish two different regimes, corresponding to $\Delta < 1$ and $\Delta > 1$ where $\Delta = \frac{1}{2}(q + q^{-1})$. Borrowing the terminology from the XXZ spin chain, we call $\Delta < 1$ and $\Delta > 1$ the massless and massive regimes, respectively. We will see that the results in the two different regimes are qualitatively different.

## 5.1 Massive regime

We first consider the massive regime which is simpler. We take $\xi = 9$ and $q = 2$. We would like to emphasize that the specific values of $\xi$ and $q$ are not crucial, as they can be easily adjusted. In practice, working with rational values (whether real or complex) is more convenient than using irrational ones for technical reasons.[9] According to (9), this corresponds to the regime where $\alpha$ and $\phi$ are complex numbers. We find that

$$\widehat{\mathcal{D}}_2^{(2)}(\omega, 0) = -\frac{156104641a^2 - 3440094482a + 10884540241}{35153041a^3 - 1005425523a^2 + 6186972723a - 10884540241}. \tag{97}$$

We can plot the absolute value $|\widehat{\mathcal{D}}_L^{(2)}(\omega, 0)|$ as a function of $\omega$, which is given in Figure 7. Taking the same parameters $\xi = 9$ and $q = 2$, we present a few more results for

$$\widehat{\mathcal{D}}_L^{(2)}(\omega, 0) = \frac{\hat{P}_L^{(2)}(a)}{\hat{Q}_L^{(2)}(a)}. \tag{98}$$

For $L = 3$, $M = 2$, we have

$$\hat{P}_3^{(2)}(a) = 25626566889a^3 + 2808199326933a^2 \\ + 88100949869467a - 1135573198803289,$$

$$\hat{Q}_3^{(2)}(a) = -208422380089a^4 + 4607080618756a^3 + 5564104859466a^2 \\ + 96035779705156a - 1135573198803289.$$

---

[9] For rational parameter values, the final results depend only on rational numbers, which simplifies the fitting process.

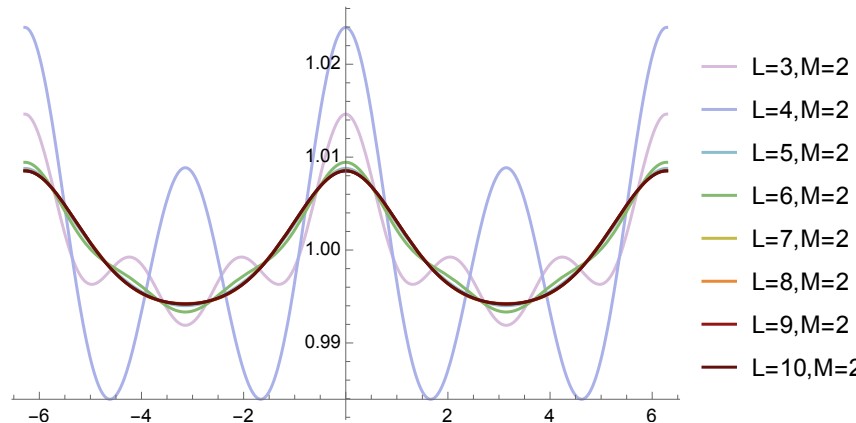

Figure 8: Plot of $|\widehat{\mathcal{D}}_L^{(2)}(\omega,0)|$ for $L = 3 \sim 10$. The curves converge to a steady shape as $L$ increases.

Table 1: Number of solutions that contribute for $L = 3 \sim 10$, $M = 2$.

| $L$ | 3 | 4 | 5 | 6 | 7 | 8 | 9 | 10 |
|---|---|---|---|---|---|---|---|---|
| $\mathcal{N}_s$ | 5 | 8 | 9 | 12 | 13 | 16 | 17 | 20 |

For $L = 4$, $M = 2$, we have

$$\hat{P}_4^{(2)}(a) = 607759660339524a^4 + 418511232917558704a^3 - 7464677848782143656a^2$$
$$- 38414127109876466896a + 47389286503179335324,$$

$$\hat{Q}_4^{(2)}(a) = -4942945166190724a^5 + 114584571922332820a^4$$
$$+ 9361729807244773560a^3 - 14807141945943110360a^2$$
$$- 41725458557586857620a + 47389286503179335324.$$

For $L = 5$, $M = 2$, we have

$$\hat{P}_5^{(2)}(a) = 900851756538259449a^5 + 295979875860427784755a^4$$
$$+ 2984695966367841261690a^3 + 13961140703024682671910a^2$$
$$+ 10449153994986136986874845a - 1236019217897549216365249,$$

$$\hat{Q}_5^{(2)}(a) = -7326680472586200649a^6 + 177733069582320607894$$
$$a^5 + 329949279946254002265a^4$$
$$+ 594108740215492080498 0a^3 + 2255645938669841730226 5a^2$$
$$+ 113128237415065803713589 4a - 1236019217897549216365249.$$

Taking $a = e^{i\omega}$, the corresponding plot is given in Figure 8. For higher quantum numbers, we can obtain similar exact results. The results for $M = 2$ and $L = 3 \sim 5$ can be found in the Appendix E. We also obtained analytic results for 4-magnon states for $L = 4,5,6,7,8$. These results except for $L = 4, M = 4$ are too bulky to be presented in the paper. Interested readers can find this data in the attached file of the arXiv submission. We present the plot for the Fourier space correlation functions.

For 2-magnon states, as $L$ increases, the analytic results become more complicated, as expected. The number of solutions that we need to take into account increases. This is given in Table 1. However, as we can see from the plot, the shape of the curve quickly converges. For large enough $L$, the difference between $|\widehat{\mathcal{D}}_L^{(2)}(\omega,0)|$ and $|\widehat{\mathcal{D}}_{L+1}^{(2)}(\omega,0)|$ become smaller. Their

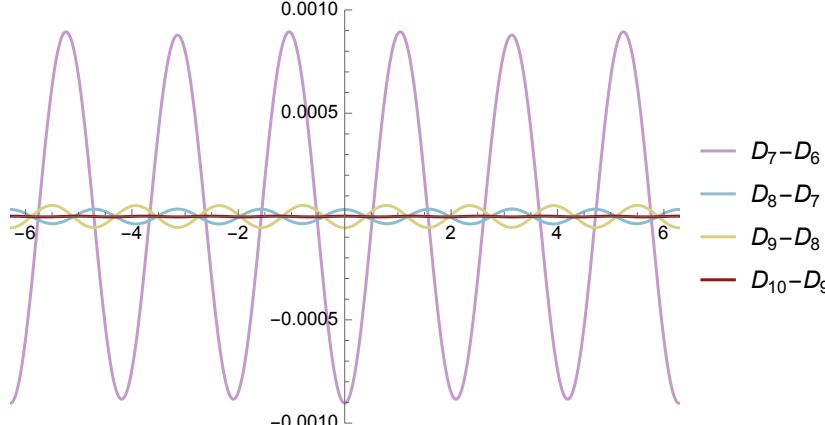

Figure 9: Plot of $|\widehat{\mathcal{D}}_{L+1}^{(2)}(\omega,0)| - |\widehat{\mathcal{D}}_L^{(2)}(\omega,0)|$ for $L = 6 \sim 9$.

Table 2: Number of Bethe roots that contributes for $L = 4 \sim 9$ and $M = 4$.

| $L$ | 4 | 5 | 6 | 7 | 8 | 9 |
|---|---|---|---|---|---|---|
| $\mathcal{N}_s$ | 20 | 42 | 85 | 143 | 232 | 340 |

difference is plotted in Figure 9. As we can see, the difference $|\widehat{\mathcal{D}}_{10}^{(2)}(\omega,0)| - |\widehat{\mathcal{D}}_9^{(2)}(\omega,0)|$ is of order $10^{-6}$. This implies that for relatively large $L$, the result is already a good approximation for the infinite length limit $L \to \infty$.

For 4-magnon states, we have the similar observation. The computation is significantly more involved. The number of Bethe roots which contributes are given in Table 2. The curves converge faster than for the 2-magnon states. The plot for $L = 4 \sim 8$ is given in Figure 10. The differences are

$$|\widehat{\mathcal{D}}_7^{(2)}(\omega,0)| - |\widehat{\mathcal{D}}_6^{(2)}(\omega,0)| \sim 10^{-5}\,, \qquad |\widehat{\mathcal{D}}_8^{(2)}(\omega,0)| - |\widehat{\mathcal{D}}_7^{(2)}(\omega,0)| \sim 10^{-8}\,. \qquad (99)$$

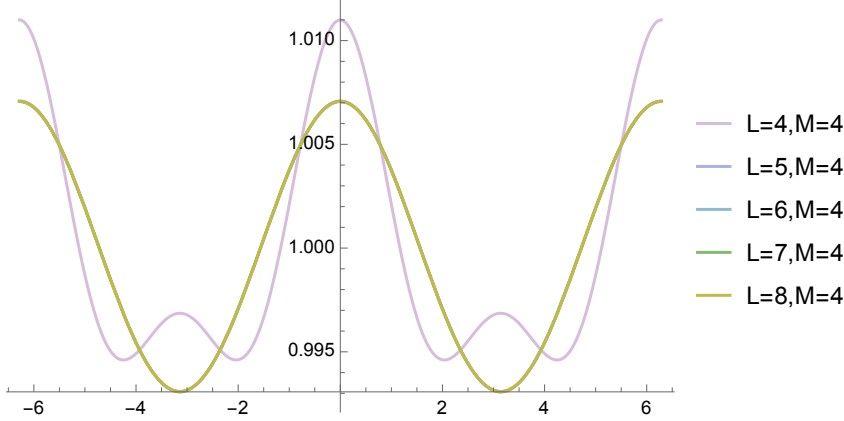

Figure 10: Plot of $|\widehat{\mathcal{D}}_L^{(4)}(\omega,0)|$ for $L = 4 \sim 8$. The curves converge so quickly and we cannot distinguish the curves for $L \geq 5$ in the figure.

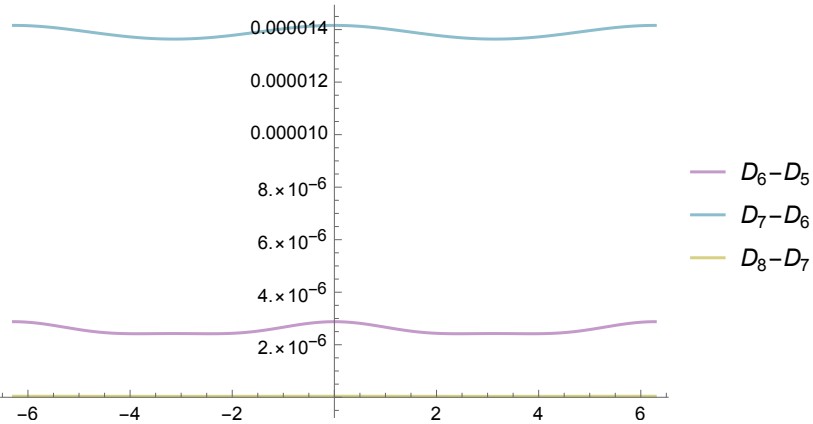

Figure 11: Plot of $|\widehat{\mathcal{D}}^{(4)}_{L+1}(\omega,0)| - |\widehat{\mathcal{D}}^{(4)}_L(\omega,0)|$ for $L = 5 \sim 7$.

They are plotted in Figure 11. Let us explain the behavior of $\widehat{\mathcal{D}}^{(N)}(\omega,Q_0)$ in the massive regime. From explicit numerical calculation, we find that in this regime the eigenvalues of $\tau(\mathbf{u}_N)$ satisfy $0 < \tau(\mathbf{u}_N) < 1$. According to (91), we can thus write approximately

$$\widehat{\mathcal{D}}^{(N)}(\omega,Q_0) \approx L \sum_{\mathrm{sol}_{N,Q_0}} w(\mathbf{u}_N)\left(1 + e^{-\mathrm{i}\omega-\gamma}\tau(\mathbf{u}_N)\right),$$

which is a periodic function. For fixed $N$, as $L$ increases, the maximal eigenvalue $\tau(\mathbf{u}_N)$ stabilize which explains the convergence of the curves in Figure 8 and 10.

## 5.2 Massless regime

In the massless regime, we take $\xi = 9$, $q = \frac{3}{5} + \frac{4\mathrm{i}}{5}$. According to (9), this corresponds to the regime where $\alpha$ and $\phi$ are real numbers. The behaviors of $|\widehat{\mathcal{D}}^{(M)}_L(\omega,0)|$ are rather different from the massive regime. For fixed $M$, with increasing $L$, the curves do not obviously converge. For $M = 2$, we plot the $|\widehat{\mathcal{D}}^{(2)}_L(\omega,0)|$ with $L = 4,5,6,7,8,9$ in Figure 12. Their comparison is given in Figure 13 The analytic expressions for $\widehat{\mathcal{D}}^{(2)}_L(\omega,0)$ can be found in the Appendix. Similarly, we have obtained the analytic results for $M = 4$ with $L = 4,5,6,7,8,9$. Their comparison is given in Figure 15. The analytic expression of the simplest one, namely $\widehat{\mathcal{D}}^{(4)}_4(\omega,0)$ are two long to be presented explicitly on the paper. Interested readers can download them from the auxiliary file. We plot the results of $\widehat{\mathcal{D}}^{(4)}_L(\omega,0)$ in Figure 14.

In the massless regime, the eigenvalues of $\tau(\mathbf{u}_N)$ satisfy $|\tau(\mathbf{u}_N)| = 1$, namely they are distributed on the unit circle. According to (91), a local maximum of $\widehat{\mathcal{D}}^{(N)}(\omega,Q_0)$ is reached when $\omega$ satisfies $e^{\mathrm{i}\omega} = \tau(\mathbf{u}_N)$, these values correspond to the location of the peaks in Figure (12) and (14). The height of each peak is characterized by the corresponding $w(\mathbf{u}_N)$. Note that $w(\mathbf{u}_N)$ may vanish for certain solutions.

## 6 Conclusions

In this paper, we developed an analytic approach to compute a class of correlation functions for a string of operators in an integrable quantum circuit. The analytic result could be used to test real-world implementations of quantum circuits. Our method is based on the Bethe Ansatz solution of the model and computational algebraic geometry, which allows us to sum over all solutions of the BAE analytically. The results we obtain are exact and analytic, without

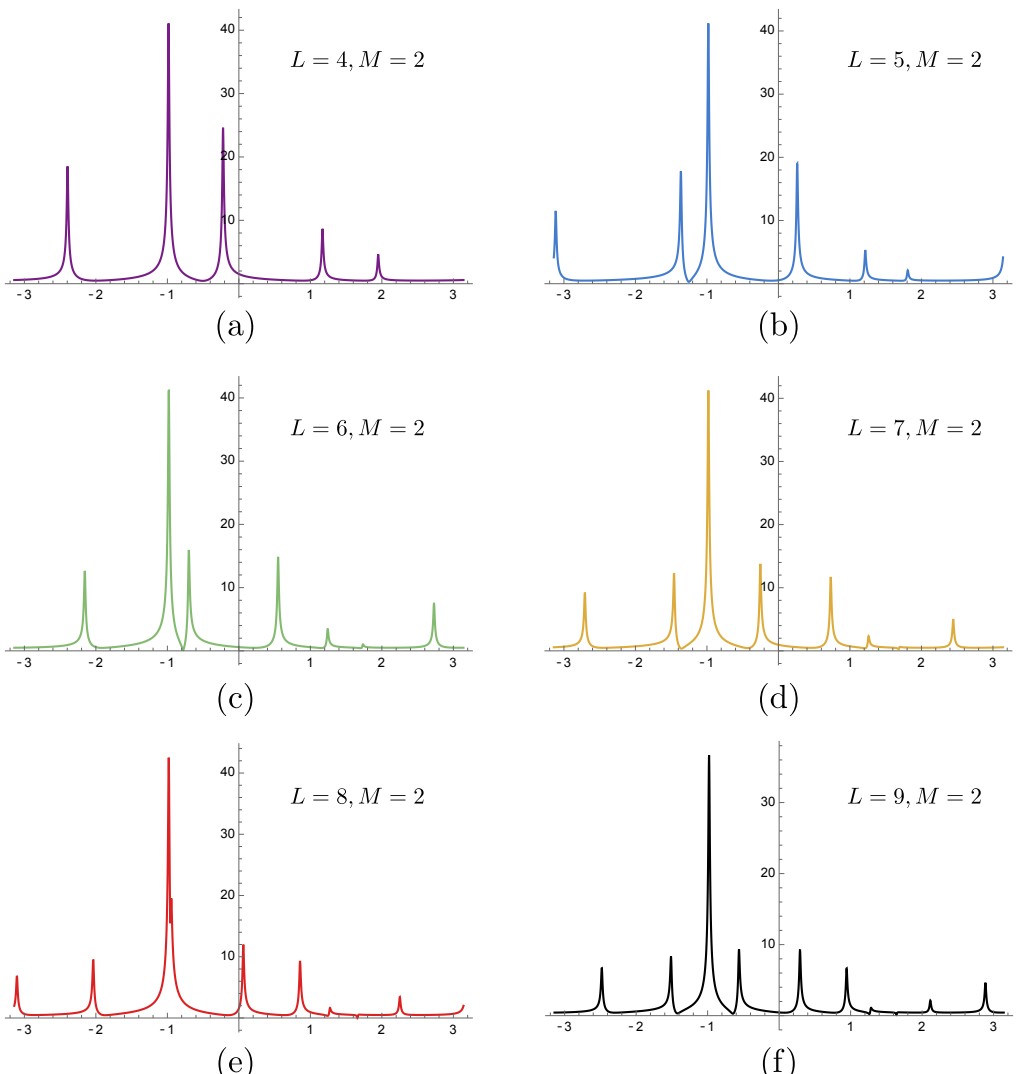

Figure 12: The plot of $|\widehat{\mathcal{D}}_L^{(2)}(\omega,0)|$ for $L = 4,5,6,7,8,9$. We take the range $-\pi \leq \omega < \pi$ with $\gamma = 0.01$.

any approximations. We obtained results for medium size systems with $10 \sim 20$ qubits, which is precisely the typical size of quantum circuits at the present stage. It is important that the quantities that we compute are accessible to present day quantum computers, therefore our results could be used as benchmarks for the quantum computing platforms.

We obtain results in both real space and Fourier space. In the real space, the results of the correlation functions are rational functions of parameters $\xi = e^u$ and $q = e^\eta$ (accordingly, in the isotropic limit it is a function of $u$ and $\eta$). Using the analytic results, we investigate both the short-time and long-time behaviors of the correlation functions.

In the short time limit, we find that the result does not depend on the size of the system if the size is larger than $2n$ where $n$ is the discrete time. This is in agreement with the light-cone propagation of information in the brickwork quantum circuits.

In the long time limit for fixed system size, the result is equivalent to a partial thermodynamic limit of the cylinder partition function of the 6-vertex model. Viewing the final result as a function of the spectral parameter, we investigate its distribution of zeros. We found that the zeros condense on a few condensation curves, similar to the zeros of the partition function of the 6-vertex model. This is a consequence of the BKW theorem. We also compared the results

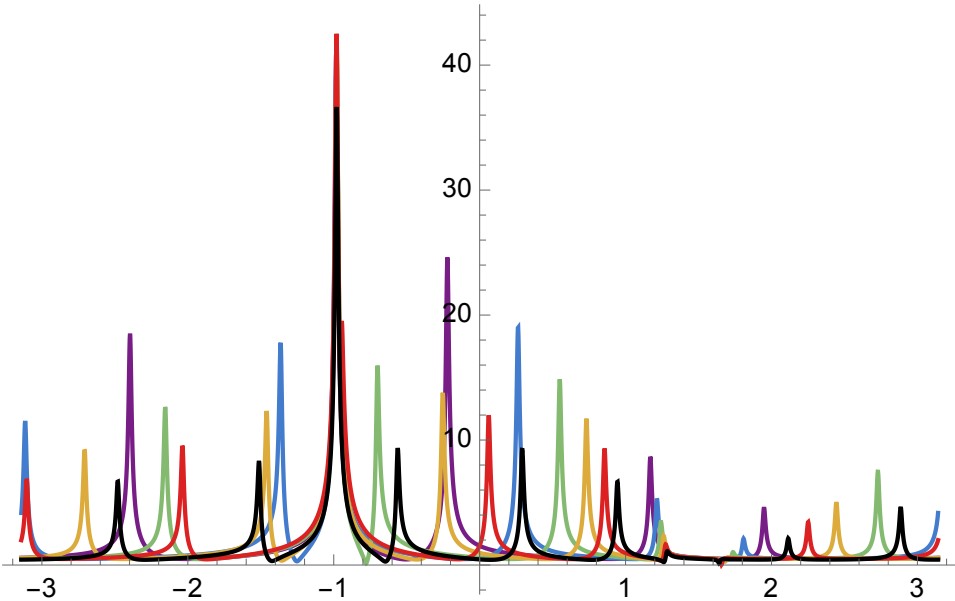

Figure 13: Comparison of $|\widehat{\mathcal{D}}_L^{(2)}(\omega, 0)|$ with $L = 4, 5, 6, 7, 8, 9$. The curves do not obviously converge as $L$ increases.

to another choice of the string of operators, which corresponds to the so-called dimer state in the XXZ spin chain. For $L = 8$, $M = 4$, we found that the condensation curves for the two results overlap in the lower half plane, but are quite different in the upper half plane. These zeros are analogous to the Lee-Yang zeros of the Loschmidt echo, which plays a central role in the context of dynamical phase transition. Such dynamical Lee-Yang zeros can be measured in experiments in a slightly different set-up, see for example [59]. It thus presents an intriguing open question and a significant challenge for measuring Lee–Yang zeros in the context of quantum circuits.

In Fourier space, the results are rational functions of $\xi$, $q$ and $a = e^{i\omega}$. We can obtain analytic results for generic fixed $\xi$ and $q$, which are rational functions in $a$. The plot of such results should be comparable to experimental data. We obtain these results for the length $L = 4 \sim 9$ (corresponding to $8 \sim 18$ qubits) with fixed parameters $\xi$ and $q$. We obtained results for both two- and four-magnon states. We investigated two sets of parameters $(\xi, q) = (9, 2), (2, 3/5 + 4i/5)$. Using the terminology from the XXZ spin chain, they are in the massive and massless regimes, respectively. Using the parametrization of the quantum circuits, they correspond to the case where $\alpha, \phi$ are complex and real. We observed that the results are qualitatively different. For the massive regime, the results are analytic for real $\omega$. For fixed magnon number $M$, the numerical results quickly converge with increasing $L$. Therefore $L = 9$ is already a very good approximation for the result in the thermodynamic limit. In the massless regime, the result as a function of $\omega$ has poles; therefore, we need to set a small non-zero $\gamma$ in $a = e^{i\omega+\gamma}$ to regularize the result. We find that for fixed magnon numbers, the results do not converge quickly, and the results are rather different for different $L$. Therefore, it is questionable whether in such a case we can apply the thermodynamic approximation for a finite $L$ result.

There are several interesting questions that one can investigate following the current work. In the current paper, we mainly computed the correlation function of a string of operators which corresponds to domain wall states. By identifying the correlation function of another string of operators with the cylinder partition function of the 6-vertex model, we find that they exhibit different behaviors in the long time limit. It would be interesting to compute the

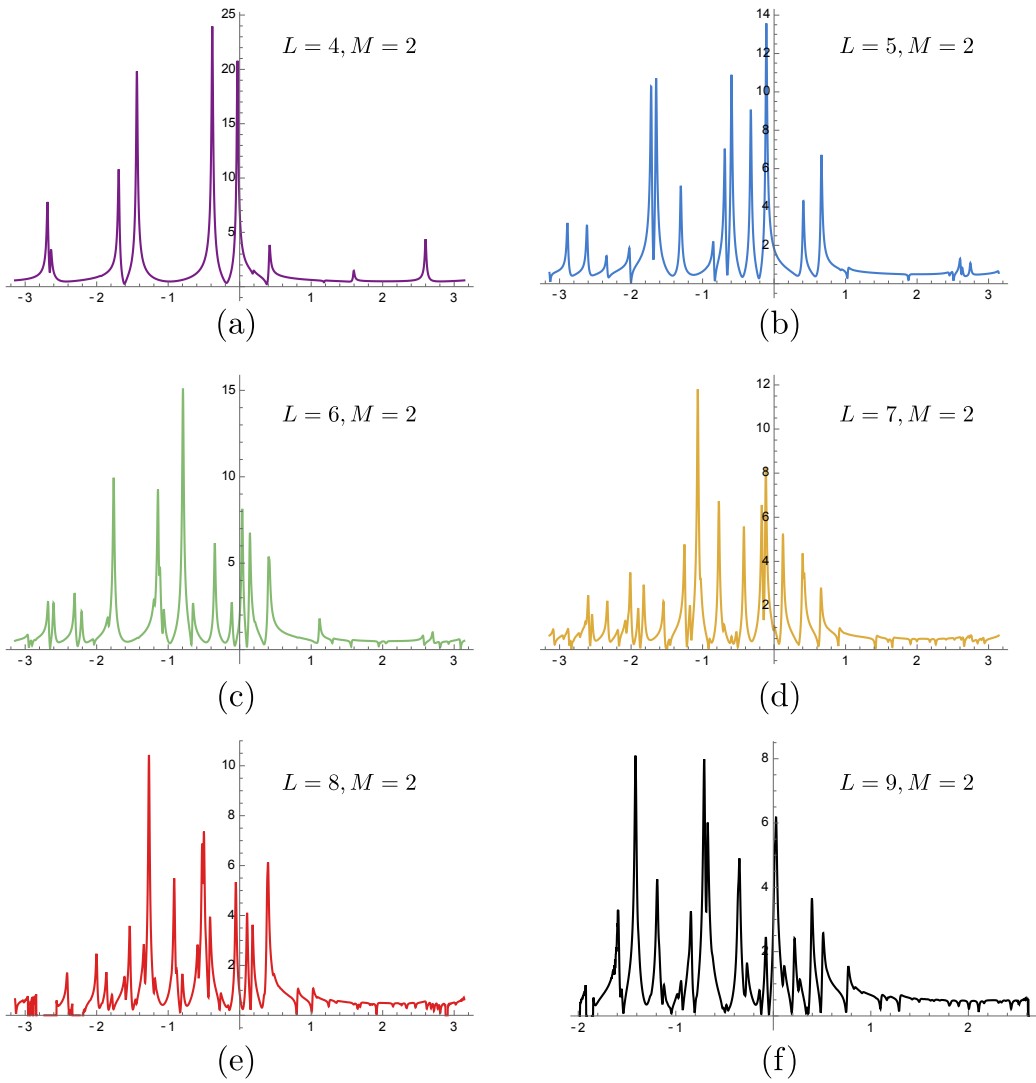

Figure 14: The plot of $|\widehat{\mathcal{D}}_L^{(4)}(\omega, 0)|$ for $L = 4, 5, 6, 7, 8$. We take the range $-\pi \le \omega < \pi$ and $\gamma = 0.01$.

correlation functions of more types of string operators and investigate universal behaviors for correlation functions in quantum integrable circuits. To address this, it is essential to derive an analytical expression for the overlap between the Bethe state and the state generated by acting string operators on the pseudovacuum. A closely related challenge, which is important for enhancing the efficiency of analytical computations, involves reformulating various physical quantities—such as overlaps and the Gaudin norm—in terms of the variables $\{c_k\}$ defined in (57), which are elementary symmetric polynomials of the Bethe roots $\{u_k\}$. The process of achieving this reformulation is notably intricate and computationally demanding. However, despite their fundamental nature, simple analytical expressions for the Gaudin norm in terms of $\{c_k\}$ remain elusive, which is an interesting question to investigate.

Another interesting problem is computing the correlators for the case where $q$ is a root of unity, *i.e.* $q = \exp(i\pi\ell_1/\ell_2)$. It is well-known that at this value, the BAE is much more subtle than the generic $q$ case. A naive numerical solution of the BAE will lead to infinitely many solutions due to the existence of certain bound states called Fabricius–McCoy strings [60, 61]. The rational $Q$-system at root of unity was constructed very recently and we now have a procedure to find all physical solutions [29]. Note that this is not only interesting

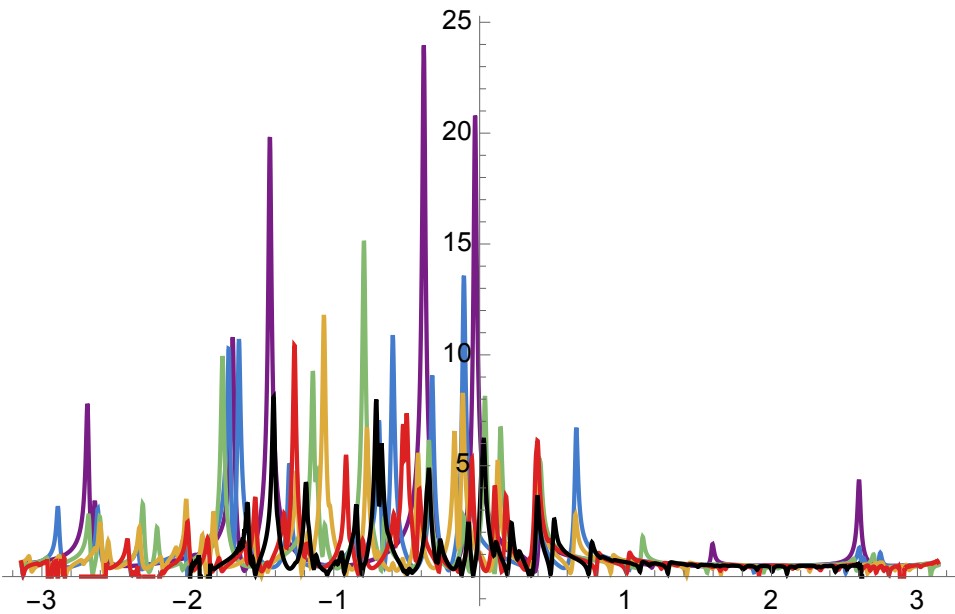

Figure 15: Comparison of $|\widehat{\mathcal{D}}_L^{(4)}(\omega, 0)|$ with $L = 4, 5, 6, 7, 8, 9$. The curves do not obviously converge as $L$ increases.

from the perspective of mathematical physics, but that physical systems at these values of parameters can exhibit novel features. One example is the spin transport property at root of unity (see *e.g.* [62] and references therein). It would be very interesting to investigate the quantum circuit model at these values.

Finally, it would be interesting to consider other integrable quantum circuit models such as the chiral Hubbard model in [18] with the techniques developed in this paper. Such models correspond to spin chains with higher rank symmetry and are technically more challenging. On the other hand, from the study of integrable models, it is expected that there would be new features such as spin charge separation. It would be fascinating to see the effect between charge and spin degrees of freedom in the quantum circuit models. Part of the theoretical tools such as the rational $Q$-system have been developed in the past years and it is now possible to study such models.

# Acknowledgments

We thank Janko Böhm, Jesper Jacobsen and Youjin Deng for the enlightening discussions, and in particular thank Yongqun Xu for remotely operating the computer cluster at USTC. The research was also supported by the advanced computing resources provided by the supercomputing center of the USTC.

**Funding information** YZ is supported by the NSF of China through Grant No. 12075234, 11947301, 12047502, and 12247103. AH acknowledges the grants PNRR MUR Project PE0000023-NQSTI and PRO3 Quantum Pathfinder.

# A  Proof of overlap formula (46)

In this appendix, we prove the formula (46). Consider the monodromy matrix with generic inhomogeneity

$$M_0(u|\{\theta_j\}) = R_{01}(u-\theta_1)R_{02}(u-\theta_2)\cdots R_{0,2L}(u-\theta_{2L}) = \begin{pmatrix} A(u) & B(u) \\ C(u) & D(u) \end{pmatrix}_0. \tag{A.1}$$

The Bethe states are generated by $|\mathbf{u}_K\rangle = B(u_1)\cdots B(u_K)|\Omega\rangle$. Taking the spectral parameter at the inhomogeneities, one notices that

$$B(\theta_{2L-N+1})\cdots B(\theta_{2L-1})B(\theta_{2L})|\Omega\rangle = |\underbrace{0\cdots0}_{2L-N}\underbrace{1\cdots1}_{N}\rangle. \tag{A.2}$$

Similarly, for the dual state we have

$$\langle\Omega|C(\theta_1)C(\theta_2)\cdots C(\theta_N) = \langle\underbrace{1\cdots1}_{N}\underbrace{0\cdots0}_{L-N}|. \tag{A.3}$$

Making the alternating choice of the inhomogeneities (24), it follows that we have

$$|0\cdots0\underbrace{1\cdots1}_{N}\rangle = B(-\tfrac{u}{2})B(\tfrac{u}{2})\cdots B(-\tfrac{u}{2})B(\tfrac{u}{2})|\Omega\rangle, \tag{A.4}$$

$$\langle\underbrace{1\cdots1}_{N}0\cdots0| = \langle\Omega|C(-\tfrac{u}{2})C(\tfrac{u}{2})\cdots C(-\tfrac{u}{2})C(\tfrac{u}{2}).$$

Using the shift operator $V$ defined in (33) and assuming that $N$ is even, we have

$$\langle\mathbf{u}_N|\mathrm{DW}_N\rangle = \langle\mathbf{u}_N|V^N|0\cdots0\underbrace{1\cdots1}_{N}\rangle \tag{A.5}$$

$$= \prod_{j=1}^{N}\left(\frac{\varphi(u_j-u/2)}{\varphi(u_k-u/2+\eta)}\frac{\varphi(u_j+u/2)}{\varphi(u_k+u/2+\eta)}\right)^{N/2}\langle\mathbf{u}_N|0\cdots0\underbrace{1\cdots1}_{N}\rangle,$$

where we have used (35) and (36). Now using the symmetric property of the scalar product (see *e.g* section 3 of [63]) we have

$$\langle\mathbf{u}_N|0\cdots0\underbrace{1\cdots1}_{N}\rangle = \langle\Omega|\prod_{j=1}^{N}C(u_j)B(-\tfrac{u}{2})B(\tfrac{u}{2})\cdots B(-\tfrac{u}{2})B(\tfrac{u}{2})|\Omega\rangle \tag{A.6}$$

$$= \langle\Omega|C(-\tfrac{u}{2})C(\tfrac{u}{2})\cdots C(-\tfrac{u}{2})C(\tfrac{u}{2})\prod_{j=1}^{N}B(u_j)|\Omega\rangle$$

$$= \langle\mathrm{DW}_N|\mathbf{u}_N\rangle.$$

Plugging (A.6) into (A.5), we obtain

$$\langle\mathbf{u}_N|\mathrm{DW}_N\rangle = \prod_{j=1}^{N}\left(\frac{\varphi(u_j-u/2)}{\varphi(u_k-u/2+\eta)}\frac{\varphi(u_j+u/2)}{\varphi(u_k+u/2+\eta)}\right)^{N/2}\langle\mathrm{DW}_N|\mathbf{u}_N\rangle, \tag{A.7}$$

which is (46) in the main text.

# B  More on computational algebraic geometry

In this appendix, we give more details for the computational algebraic geometry method (or commutative algebra) and its implementations. For a more thorough introduction to the subject, we refer to the text book [53].

## B.1 Polynomial ring and ideal

Let us consider a polynomial ring $\mathcal{A} = \mathbb{C}[x_1, \ldots, x_n]$ which consists of all polynomials in variables $\{x_k\}$ with complex coefficients. It is also possible to consider the polynomial ring over other fields instead of $\mathbb{C}$.

An *ideal* $\mathcal{I}$ of the ring $\mathcal{A}$ is a subset such that

1. If $f_1 \in \mathcal{I}$ and $f_2 \in \mathcal{I}$, then $f_1 + f_2 \in \mathcal{I}$.

2. If $f \in \mathcal{I}$, then $f g \in \mathcal{I}$ for any $g \in \mathcal{A}$.

A polynomial ring is a *Noetherian* ring, which means any ideal $\mathcal{I}$ of $\mathcal{A}$ is *finitely generated*. This means for an ideal $\mathcal{I}$, there exists a finite number of polynomials $f_i \in \mathcal{I}$ such that any polynomial $F \in \mathcal{I}$ can be expressed as

$$ F = \sum_{i=1}^{K} g_i f_i , \qquad g_i \in \mathcal{A} . \tag{B.1} $$

These polynomials $\{f_1, \ldots, f_K\}$ are called a basis for the ideal $\mathcal{I}$ and we can write $\mathcal{I} = \langle f_1, \ldots, f_K \rangle$. The ideal is intimately related to the set of polynomial equations $f_1 = f_2 = \ldots f_K = 0$. In fact, we can see that if this set of equations is satisfied, all elements in the ideal $\mathcal{I}$ vanish. The basis for a given ideal $\mathcal{I}$ is not unique and we can make a 'change of basis' to a different set of polynomials

$$ \mathcal{I} = \langle f_1, \ldots, f_K \rangle = \langle g_1, \ldots, g_{K'} \rangle , \tag{B.2} $$

where in general $K \neq K'$.

## B.2 Gröbner basis and quotient ring

Given an ideal $\mathcal{I}$, we can define the quotient ring $V_f = \mathcal{A}/\mathcal{I}$ as the quotient set specified by the following equivalence relation

$$ F \sim G , \qquad \text{if and only if} \qquad F - G \in \mathcal{I} . \tag{B.3} $$

This means for a given basis of the ideal $\mathcal{I} = \langle f_1, \ldots, f_k \rangle$ we make the following identification

$$ F \sim F + \sum_{i=1}^{K} a_i f_i , \qquad a_k \in \mathcal{A} . \tag{B.4} $$

Given a polynomial $F(u)$, we can perform a polynomial reduction with respect to $f_i$ and find a remainder. Roughly speaking, the quotient ring is the space of such remainders. However, for an arbitrary generating set, the aforementioned polynomial reduction is ill-defined because the 'remainder' is not unique. Therefore, in order to make the reduction in a polynomial ring possible, one needs a "convenient" basis such that the polynomial reduction leads to a well-defined and unique remainder. Such a convenient basis does indeed exist and is called a Gröbner basis of the ideal.

To define a Gröbner basis, we need to first define monomial orders in a polynomial ring. A monomial order $\prec$ is a total order for all monomials in $\mathcal{A}$ such that

- If $u \prec v$ then for any monomial $w$, $uw \prec vw$.

- If $u$ is a non-constant monomial, then $1 \prec u$.

In most of our calculations, we choose the *degrevlex* (DegreeReverseLexicographic) ordering. After defining a monomial order $\prec$, for any polynomial $F \in \mathcal{A}$, there is a unique *leading term*, denoted by $\mathrm{LT}(f)$ which is the highest monomial of $F$ in the order $\prec$. Now we can give the formal definition of a Gröbner basis.

**Gröbner basis** A Gröbner basis $G(\mathcal{I})$ of an ideal $\mathcal{I}$ with respect to a monomial order $\prec$ is a generating set of $\mathcal{I}$ such that for any $F \in \mathcal{I}$,

$$\exists g_i \in G(\mathcal{I}), \qquad \mathrm{LT}(g_i)|\mathrm{LT}(F), \tag{B.5}$$

where $a|b$ means a monomial $b$ is divisible by another monomial $a$. Given a monomial order $\prec$ and any generating set $f_i$ of the ideal $\mathcal{I}$, there are standard algorithms such as the Buchberger algorithm or the F4/F5 algorithm to compute the Gröbner basis. Such algorithms have been implemented in many commonly used softwares such as `Mathematica` or more specialized packages such as `Singular`. For example, we can use the `Mathematica` function `GroebnerBasis` to compute the Gröbner basis of a given ideal.

The property (B.5) ensures that the polynomial division of any polynomial $F \in \mathcal{A}$ by an ideal $\mathcal{I}$ in the order $\prec$ is well-defined

$$F = \sum a_i g_i + r, \tag{B.6}$$

where $g_i$'s are the elements of the Gröbner basis and $r$ is called the remainder, which contains monomials *not* divisible by any $\mathrm{LT}(g_i)$. The key point is that, given the monomial order $\prec$, the remainder $r$ for $F$ is unique.

**Quotient ring basis** With a Gröbner basis, the polynomial reduction now can be performed. The quotient ring $V_f = \mathcal{A}/\mathcal{I}$ can now be identified with the space of 'remainders'. A canonical basis for the quotient ring can be chosen as the monomials which are not divisible by any element in $\mathrm{LT}(G(\mathcal{I}))$. Let us denote this basis by $B = \{b_1, b_2, \ldots, b_{N_s}\}$. One important result from algebraic geometry is that the number of solutions of the equation $f_1 = f_2 = \ldots = f_K = 0$ equals the dimension of the quotient ring $\dim V_f = N_s$. This statement provides a valuable method for determining the number of solutions. In practice, we can use the lattice algorithm to list these monomials.

### B.3 Companion matrix

Let $B = \{b_1, \ldots, b_{N_s}\}$ be the canonical basis of $V_f$, for any $F \in \mathcal{A}$, we can define

$$F b_i \sim \sum_{j=1}^{N_s} b_j (\mathbf{M}_F)_{ji}, \tag{B.7}$$

where $\sim$ stands for the procedure of performing polynomial reduction with respect to the Gröbner basis and finding the remainder. $\mathbf{M}_F$ is an $N_s \times N_s$ matrix which is called the companion matrix of $F$. Let us denote the solutions of the equations $f_1 = f_2 = \ldots = f_K = 0$ by $\mathcal{Z}(I)$. We have the following important result

$$\sum_{p \in \mathcal{Z}(I)} f(p) = \mathrm{tr}(\mathbf{M}_F). \tag{B.8}$$

This is the key formula for our purpose. Note that to calculate the right hand side, we only need to compute $G(\mathcal{I})$ which is purely algebraic and analytic. The companion matrices have the following properties

$$\mathbf{M}_{F_1+F_2} = \mathbf{M}_{F_1} + \mathbf{M}_{F_2}, \qquad \mathbf{M}_{F_1 F_2} = \mathbf{M}_{F_1}\mathbf{M}_{F_2} = \mathbf{M}_{F_2}\mathbf{M}_{F_1}, \tag{B.9}$$

and also

$$\mathbf{M}_{F_1/F_2} = \mathbf{M}_{F_1} \cdot \mathbf{M}_{F_2}^{-1}. \tag{B.10}$$

This last property allows us to compute the companion matrix of a rational function $f/g$ as

$$\sum_{p \in \mathcal{Z}(I)} \frac{f(p)}{g(p)} = \text{tr}(M_f M_g^{-1}), \tag{B.11}$$

as long as $g(p) \neq 0$, $\forall p \in \mathcal{Z}(\mathcal{I})$.

### B.4 Efficient implementations

In the computational algebraic geometry approach, the majority of the computation time is spent on the computation of the Gröbner basis $G(I)$ and the companion matrix [30]. In this work, the former computation is powered by the Buchberger algorithm implemented in SINGULAR [54] and also the new parallel Buchberger algorithm implemented with SINGULAR/GPI-SPACE framework [64] [65]. The computation of the companion matrix is powered by the multivariate polynomial division function in SINGULAR. In practice, the coefficients of the equations under consideration are not rational or complex numbers, but rational functions of some physical parameters. The Gröbner basis computation with parameters is possible but inefficient. We carry out the Gröbner basis computation with integer-valued parameters for better performance. The same computation with different values of parameters can be performed in parallelization. The final physical results will come from the interpolation of the different parameters. In general, the efficiency of such interpolation depends on our knowledge about the structure of the final result.

In the context of this work, the results are presented in the form of rational functions. For the XXX model, the results are univariate rational functions. For the XXZ model, they are multivariate (bivariate) rational functions. To finish Gröbner bases computation with higher efficiency, one can perform integer-value seeding over these variables on the quotient ring representation, which will give a large amount of simpler CAG results with fixed values of variables.

Then to get the full analytic result in the form of rational functions, one can perform Thiele's rational interpolation formula for dense rational reconstruction [66].

This algorithm is supremely efficient for univariate interpolation of rational functions and is sensitive to the number of numerical points as probes in the computation. This process is finished with private implementation in SINGULAR and MATHEMATICA.

## C  Rational $Q$-system

In this appendix, we introduce the rational $Q$-system. We present the construction for the XXZ spin chain. The rational $Q$-system for XXX is similar.

**Young diagram**  We start with a two-row Young diagram with boxes $(L-M, M)$ where $M$ is the number of magnons and $L$ is the length of the spin chain. We consider the regime $M \leq L/2$. To each node of the Young diagram, we associate a $Q$-function denoted by $Q_{a,s}(x)$ where $a$ and $s$ are the lattice coordinates. For the XXZ spin chain, we consider the $Q$-functions as functions of the multiplicative variable $x = e^{2u}$.

**QQ-relations**  The $Q$-functions around each box whose lower-left corner coordinate is $(a, s)$ are related by the $QQ$-relations

$$Q_{a+1,s} Q_{a,s+1} = Q^+_{a+1,s+1} Q^-_{a,s} - Q^-_{a+1,s+1} Q^+_{a,s}, \tag{C.1}$$

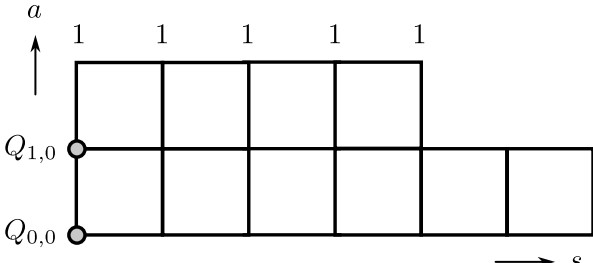

Figure 16: The Young diagram of rational $Q$-system. The bottom row has $L - M$ boxes and the top row has $M$ boxes.

where

$$Q_{a,s}^{\pm}(x) = Q_{a,s}(xq^{\pm 1}), \tag{C.2}$$

with $q = e^{\eta}$. Among all the $Q$-functions, the $Q_{1,0}(x)$ is special because its zeros are the Bethe roots. We stress that the $QQ$-relation is defined up to proportionality that is independent of the spectral parameter. So the global normalizations of the $Q$-functions are not important.

The $QQ$-relation clearly states that the $Q$-functions at different nodes are not independent. As a result, we can work out all the $Q$-functions starting with a few fixed $Q$-functions. These fixed ones are the boundary conditions.

**Boundary conditions**  We fix the $Q$-functions at the upper and left boundary. The $Q$-functions at the upper boundary are fixed to be a constant $Q_{2,s}(x) = 1$. The other two $Q$-functions at the left boundary are given by

$$Q_{0,0}(x) = (x^{1/2} - x^{-1/2})^L, \qquad Q_{1,0}(x) = \prod_{j=1}^{M} \left( (x/x_j)^{1/2} - (x_j/x)^{1/2} \right), \tag{C.3}$$

where $\{x_j\}$ are the Bethe roots. Since the global normalization is not important, we have the following parametrization

$$Q_{1,0}(x) \propto \widetilde{Q}_{1,0}(x) = \prod_{j=1}^{M} (x - x_j) = x^M + \sum_{k=1}^{M-1} c_k x^k. \tag{C.4}$$

**Zero remainder condition**  To find the Bethe roots, we need to find $Q_{1,0}$, or equivalently the coefficients $\{c_k\}$. This is achieved by the zero remainder conditions. By fixing the boundary conditions $Q_{2,0} = 1$ and (C.3), we can work out the rest of the $Q$-functions by the $QQ$-relations. However, for a generic choice of $\{c_k\}$, the resulting $Q$-functions are not Laurent polynomials in $x$ (up to an unimportant global factor). By *requiring* all the $Q$-functions to be Laurent polynomials in $x$, we obtain further constraints on $\{c_j\}$, which takes the form of a set of algebraic equations. These equations are called the *zero remainder conditions*.

One important point about the zero remainder condition is that, in fact we only need to require the $Q$-functions of the first two columns to be Laurent polynomials. More precisely, we only need to make sure that $Q_{0,1}$ and $Q_{0,2}$ are polynomials. It can be proven that this is sufficient to guarantee that the rest of the $Q$-functions are Laurent polynomials.

# D  Physical quantities in $\{c_k\}$

In this appendix, we present more details for the change of variables from $\{x_k\}$ to $\{c_k\}$. In general, we consider an expression of the form $N(\{x_j\})/D(\{x_j\})$ where $N(\{x_j\})$ and $D(\{x_j\})$

are polynomials in $\{x_j\}$. We first establish the relation between $\{x_j\}$ and $\{c_j\}$ by identifying

$$\prod_{j=1}^{M}(x - x_j) = x^M + \sum_{k=1}^{M-1} c_k x^k. \tag{D.1}$$

These lead to

$$c_{M-1} = -(x_1 + x_2 + \ldots + x_M), \tag{D.2}$$
$$c_{M-2} = x_1 x_2 + x_1 x_3 + \ldots + x_{M-1} x_M,$$
$$\vdots$$
$$c_0 = (-1)^M x_1 x_2 \cdots x_M.$$

We then introduce a new variable $w$ and an equation

$$w = \frac{N(\{x_j\})}{D(\{x_j\})} \quad \Leftrightarrow \quad N(\{x_j\}) = w\, D(\{x_j\}). \tag{D.3}$$

Combing the two sets of equations (D.2) and (D.3), we obtain a system of algebraic equations involving $\{w, x_1, \ldots, x_M, c_0, \ldots, c_{M-1}\}$. We can then compute the Gröbner basis of this set of algebraic equations. The key observation is that, among the Gröbner basis, there is always a polynomial that involves only $\{w, c_0, \ldots, c_{M-1}\}$ and the dependence on $w$ is linear. Let us denote this polynomial by $S(w, c_0, \ldots, c_{M-1})$. We need to solve the equation

$$S(w, c_0, \ldots, c_{M-1}) = 0, \tag{D.4}$$

for $w$. Since the dependence on $w$ is linear, it can be solve readily and obtain $w = \widetilde{S}(\{c_k\})$. The quantity $\widetilde{S}(\{c_k\})$ is the result of symmetrization of $N(\{x_j\})/D(\{x_j\})$, *i.e.*

$$\widetilde{S}(\{c_k\}) = \frac{N(\{x_j\})}{D(\{x_j\})}. \tag{D.5}$$

## E  More analytic results in massive regime

In this appendix, we present analytic results for $L = 6, \ldots, 10$ and $M = 2$ and $L = 8$, $M = 4$ in Fourier space. We fix $\xi = 9$, $q = 2$ as in the main text.

**Result for $L = 6, M = 2$**   For $L = 6$, $M = 2$, we have

$$\begin{aligned}
\hat{P}_6^{(2)}(a) = {}& -5341150064515340273121\, a^6 - 169734594354674880220074\, a^5 \\
& - 6772978257133986995968615\, a^4 + 1208144728170975147670407220\, a^3 \\
& + 1370616640991778706771271985\, a^2 + 1134999293409968790209917517526\, a \\
& - 1289526489840334121941717217521,
\end{aligned}$$

$$\begin{aligned}
\hat{Q}_6^{(2)}(a) = {}& 43439888521963583647921\, a^7 - 1100559700418854159130647\, a^6 \\
& - 17684388051998825832880059\, a^5 - 1516485021872139366677235\, a^4 \\
& + 239719799023112498118956235\, a^3 + 2236019003611924904658601659\, a^2 \\
& + 122510509439470012808294498647\, a \\
& - 1289526489840334121941717217521.
\end{aligned}$$

**Result for $L = 7, M = 2$** For $L = 7$, $M = 2$, we have

$\hat{P}_7^{(2)}(a)$

$= -316676787325114524793344409\,a^7 - 9722535487280816894029475337\,a^6$

$- 13284014090995839018704234589\,a^5 - 2658877056547083933342929909685\,a^4$

$- 7056089646281879410086216120315\,a^3 - 132402145030090476007173235599411\,a^2$

$- 12309244542150474755821891018602463\,a$

$+ 134535009158552218608057415586748409\,,$

$\hat{Q}_7^{(2)}(a)$

$= 257555099046722087448523609\,a^8 - 6802579045483603414528506472\,a^7$

$- 9072737719011528891925528548\,a^6 - 192957938169692945243642474904\,a^5$

$- 5305531949916488629095660263370\,a^4 - 10904271570463836869480995222104\,a^3$

$- 219419179113019027067093413101348\,a^2$

$- 13249309353244078330717402870175272\,a$

$+ 134535009158552218608057415586748409\,.$

**Result for $L = 8, M = 2$** For $L = 8$, $M = 2$, we have

$$\hat{P}_8^{(2)}(a) = -187757667205060401749973710961\,a^8$$
$$- 5562295426349338066893801548712\,a^7$$
$$- 68460063735650825294029417860108\,a^6$$
$$- 43586334407867250947331928333142584\,a^5$$
$$+ 777575727956544615801669787675266730\,a^4$$
$$+ 626223846698148298352660070921579816\,a^3$$
$$+ 12538451422031669290868599967103417492\,a^2$$
$$+ 1333027799413296620840891103020100760888\,a$$
$$- 14035902970502594415160022110749874762561\,,$$

$$\hat{Q}_8^{(2)}(a) = 1527044182248015256482296477761\,a^9$$
$$- 41976962049572871860538871397449\,a^8$$
$$- 43647594567560624615959514672604\,a^7$$
$$- 950080126167093004906431342935124\,a^6$$
$$- 97727582196713429284273985905005314\,a^5$$
$$+ 1543505891415224072829997935348218114\,a^4$$
$$+ 973825436919024486110553161407549524\,a^3$$
$$+ 21275744879111919409721635343616176604\,a^2$$
$$+ 1431103821089881188206164958982840351049\,a$$
$$- 14035902970502594415160022110749874762561\,.$$

**Result for $L = 9, M = 2$**    For $L = 9$, $M = 2$, we have

$$
\begin{aligned}
\hat{P}_9^{(2)}(a) = {}&- 1113215208858803121975594132287769\, a^9 \\
&- 3178003030481669731829561847561 6879\, a^8 \\
&- 346116949924399518400809244763290884\, a^7 \\
&- 3421179821316113328851446955867198604\, a^6 \\
&- 2172371663730681721093427279453411135694\, a^5 \\
&- 2857540521689186032968321538208467721106\, a^4 \\
&- 5316887684082832194640351257197721 9887796\, a^3 \\
&- 1157398391847265559110526743401755305933116\, a^2 \\
&- 1441664470146331831544206339527319 6460154721\, a \\
&+ 146435172100956517273922994679242368 4103226569\,,
\end{aligned}
$$

$$
\begin{aligned}
\hat{Q}_9^{(2)}(a) = {}&9053844956548482455683535816644969\, a^{10} \\
&- 25863147589220913886360935335196 8890\, a^9 \\
&- 1881390263387315236046788977711 21195\, a^8 \\
&- 433521491478900802906744982804 0746680\, a^7 \\
&- 10876706082743876027814524334834 9756510\, a^6 \\
&- 434236026503703482150035762623 9933445788\, a^5 \\
&- 38337318903363426225415145942114 9169310\, a^4 \\
&- 839368105227290900253961869678 17696823480\, a^3 \\
&- 203336416088661204327063902472 6458464777995\, a^2 \\
&- 1543986202801295744830722900720 09855162061690\, a \\
&+ 146435172100956517273922994679242368 4103226569\,.
\end{aligned}
$$

**Result for $L = 10, M = 2$**    For $L = 10$, $M = 2$, we have

$$
\begin{aligned}
\hat{P}_{10}^{(2)}(a) = {}&- 66002529733238437101932976103341 82401\, a^{10} \\
&- 181316000177945635411970895396128517990\, a^9 \\
&- 170532974020062095653446525465 9507286845\, a^8 \\
&- 1075433269342298729690507924968 0362609480\, a^7 \\
&- 2788597865691928377096693175076 2365032046610\, a^6 \\
&+ 497564943588073965118725621401 742897822373852\, a^5 \\
&+ 200286748319860161399864219482 404031581811790\, a^4 \\
&+ 421182492891255112721375096154 3388449686740920\, a^3 \\
&+ 1031771406405548051786211337522 36693674989553555\, a^2 \\
&+ 1557208777609362747062310217030 0915263222808880410\, a \\
&- 1527743507012069249067111211189067 70538805524717201\,,
\end{aligned}
$$

$$\hat{Q}_{10}^{(2)}(a) = 53680246747375952479747683856888021201\,a^{11}$$
$$- 15912341731457367716434104837193951052115\,a^{10}$$
$$- 63435340665213764511102884788995900994 5\,a^{9}$$
$$- 17199644191884346788964056757783189794165\,a^{8}$$
$$- 47798624072841040971494174669641817405967 0\,a^{7}$$
$$- 6264172348173838582350446246986663338821086 2\,a^{6}$$
$$+ 9882116944059381271472928851477938271489628 62\,a^{5}$$
$$+ 2526852920161237197371415082617958752163796 70\,a^{4}$$
$$+ 68612299556457711699241392910850121675755941 65\,a^{3}$$
$$+ 190852983845750775085500945828236892588740889945\,a^{2}$$
$$+ 16639600180709600481550008015125921289340610492 11\,a$$
$$- 1527743507012069249067111211189067705388055247172 01\,.$$

We also present the result for $L = 4$, $M = 4$. As we can see, the result is considerably more complicated.

**Result for $L = 4, M = 4$**

$$\hat{P}_4^{(4)}(a) = -815617213253176171698092052909590125925465004256208 1\,a^{11}$$
$$- 20998071783853815347919633492950085709669367907434150 9\,a^{10}$$
$$+ 9805779563942996214406401795772195686233369021819555215945\,a^{9}$$
$$- 48640942527280152103556145542464029569995839671887109283223 5\,a^{8}$$
$$- 19398767636794702594635762313443606166288950348747029297528330\,a^{7}$$
$$+ 13794994538385112608321842103956151098630350107524814001525982 2\,a^{6}$$
$$+ 10499080968484130391123037715450153560066110448800564561167377 8\,a^{5}$$
$$- 1067502230719708323373110373611855273607688310911791538353875523670\,a^{4}$$
$$+ 1375792062432205164676803859076610041549655222286307457958579784223 5\,a^{3}$$
$$+ 1402067887260469722185156109817321641739101592462243378011020598860 55\,a^{2}$$
$$- 362405203824894683955033973028680510437736739186394290704287512112129\,a$$
$$+ 1809966868929172933994874978249710316608864595987799100807916546403128\,,$$

$$\hat{Q}_4^{(4)}(a) = 6633462904496682471876526449521206936367739383038488 1\,a^{12}$$
$$- 2764312726378830960965564181491288170998205011881818817 2\,a^{11}$$
$$+ 1360610363855872619271697002210675787787122935496392344546\,a^{10}$$
$$+ 738452372260002794215253387919211881384713064762005125501 80\,a^{9}$$
$$- 4780979055224263220320109492155092475298551712325064333203905\,a^{8}$$
$$- 3756254560654065111410131207147041897968983231221788166624975 2\,a^{7}$$
$$+ 52146419408318118981962926221702022010059284967072872488661932 44\,a^{6}$$
$$- 42518415110005549339013514206450781815189905693149656045060534552\,a^{5}$$
$$- 1658663539373636328709728248947847374053887035751846492027459307905\,a^{4}$$
$$+ 26300083629274148442984818223919740731189116203000185638476277686180\,a^{3}$$

$$+ 95689865821052118423630225085462686944116781710236380669755943826146\, a^2$$
$$- 37505236662190574577755948232277935947115853265201847520225999495 3217\, a$$
$$+ 18099668689291729339948749782497103166088645959877991008079165464 03128\,.$$

## F  More analytic results in massless regime

In this appendix, we present analytic results for $L = 4, \ldots, 8$ and $M = 2$ and $L = 8$, $M = 4$ in Fourier space. We fix $\xi = 9$, $q = \frac{3}{5} + \frac{4i}{5}$ as in the main text.

**Result for $L = 4, M = 2$**

$$
\begin{aligned}
\hat{P}_4^{(2)}(a) = {} & (16190945280 + 23440752896\,\mathrm{i})a^4 + (3909369240 - 18626610017\,\mathrm{i})a^3 \\
& + (18961677720 + 16502505825\,\mathrm{i})a^2 - (32861123280 + 17495261183\,\mathrm{i})a \\
& + (12788123040 + 51977892479\,\mathrm{i})\,, \\
\hat{Q}_4^{(2)}(a) = {} & (12788123040 - 51977892479\,\mathrm{i})a^5 - (16670178000 - 40936014079\,\mathrm{i})a^4 \\
& + (22871046960 - 35129115842\,\mathrm{i})a^3 + (22871046960 + 35129115842\,\mathrm{i})a^2 \\
& - (16670178000 + 40936014079\,\mathrm{i})a + (12788123040 + 51977892479\,\mathrm{i})\,.
\end{aligned}
$$

**Result for $L = 5, M = 2$**

$$
\begin{aligned}
\hat{P}_5^{(2)}(a) = {} & -(13306340474624 + 3273759498240\,\mathrm{i})a^5 \\
& + (3863467459072 + 13489014988800\,\mathrm{i})a^4 \\
& + (7773196913152 - 3163367669760\,\mathrm{i})a^3 \\
& - (6745452149248 + 12433548165120\,\mathrm{i})a^2 \\
& + (10796522125247 + 19922035736520\,\mathrm{i})a \\
& + (11977223406401 - 22791452542200\,\mathrm{i})\,, \\
\hat{Q}_5^{(2)}(a) = {} & (11977223406401 + 22791452542200\,\mathrm{i})a^6 \\
& - (2509818349377 + 23195795234760\,\mathrm{i})a^5 \\
& - (2881984690176 - 25922563153920\,\mathrm{i})a^4 \\
& + 15546393826304 a^3 \\
& - (2881984690176 + 25922563153920\,\mathrm{i})a^2 \\
& - (2509818349377 - 23195795234760\,\mathrm{i})a \\
& + (11977223406401 - 22791452542200\,\mathrm{i})\,.
\end{aligned}
$$

**Result for $L = 6, M = 2$**

$$
\begin{aligned}
\hat{P}_6^{(2)} = {} & (3066169192038656 + 5834611850803200\,\mathrm{i})a^6 \\
& + (4787760828350976 - 6497951173079040\,\mathrm{i})a^5 \\
& - (2879859037024545 + 105889836591000\,\mathrm{i})a^4 \\
& + (2599750449819425 - 2871676258248600\,\mathrm{i})a^3 \\
& - (7491925945812480 - 133968233779200\,\mathrm{i})a^2 \\
& + (13197453475178049 + 1833272092202400\,\mathrm{i})a \\
& - (4384188648550081 + 11582273787266160\,\mathrm{i})\,,
\end{aligned}
$$

$$\hat{Q}_6^{(2)} = (4384188648550081 - 11582273787266160\,\mathrm{i})a^7$$
$$- (10131284283139393 - 7667883943005600\,\mathrm{i})a^6$$
$$+ (12279686774163456 - 6363982939299840\,\mathrm{i})a^5$$
$$- (5479609486843970 + 2977566094839600\,\mathrm{i})a^4$$
$$+ (5479609486843970 - 2977566094839600\,\mathrm{i})a^3$$
$$- (12279686774163456 + 6363982939299840\,\mathrm{i})a^2$$
$$+ (10131284283139393 + 7667883943005600\,\mathrm{i})a$$
$$- (4384188648550081 + 11582273787266160\,\mathrm{i}).$$

**Result for $L = 7, M = 2$**

$$\hat{P}_7^{(2)} = - (2965062089540136960 + 1122352294028820736\,\mathrm{i})a^7$$
$$+ (186664534285271040 + 4527404219884142080\,\mathrm{i})a^6$$
$$+ (1883104092849070080 - 1055138737458676224\,\mathrm{i})a^5$$
$$- (287336924931686400 + 531319906398277120\,\mathrm{i})a^4$$
$$- (2502921301220966400 - 653553357493434880\,\mathrm{i})a^3$$
$$+ (3740222195766927360 + 1789593583871064576\,\mathrm{i})a^2$$
$$- (5298184152707352840 + 5464802435030160895\,\mathrm{i})a$$
$$+ (-2116437424659875880 + 5568174742303293439\,\mathrm{i}),$$
$$\hat{Q}_7^{(2)} = (2116437424659875880 + 5568174742303293439\,\mathrm{i})a^8$$

$$+ (2333122063167215880 - 6587154729058981631\,\mathrm{i})a^7$$
$$- (3553557661481656320 - 6316997803755206656\,\mathrm{i})a^6$$
$$+ (4386025394070036480 - 401585379965241344\,\mathrm{i})a^5$$
$$- (1062639812796554240\,\mathrm{i})a^4$$
$$- (4386025394070036480 + 401585379965241344\,\mathrm{i})a^3$$
$$+ (3553557661481656320 + 6316997803755206656\,\mathrm{i})a^2$$
$$- (2333122063167215880 + 6587154729058981631\,\mathrm{i})a$$
$$- (2116437424659875880 + 5568174742303293439\,\mathrm{i}).$$

**Result for $L = 8, M = 2$**

$$\hat{P}_8^{(2)} = (541807980712928225280 + 1425452734029643120384\,\mathrm{i})a^8$$
$$+ (1860089695679579811840 - 1663719405239146709504\,\mathrm{i})a^7$$
$$- (1306086559669141401600 + 583178930942280794624\,\mathrm{i})a^6$$
$$+ (39299130340212338280 + 290838133169585319649\,\mathrm{i})a^5$$
$$- (283996239369855701400 + 243181411869805030625\,\mathrm{i})a^4$$
$$- (555465568612198625280 - 1485219409999454373376\,\mathrm{i})a^3$$
$$+ (192621106131864016896 0 - 1440065866208923122176\,\mathrm{i})a^2$$
$$- (4232671666695290048400 - 381810663241616919679\,\mathrm{i})a$$
$$(1329399368762685232320 + 2538165215672305923841\,\mathrm{i}),$$

$$\hat{Q}_8^{(2)} = (1329399368762685232320 - 2538165215672305923841\,\mathrm{i})a^9$$
$$- (3690863685982361823120 - 1043642070788026200705\,\mathrm{i})a^8$$
$$+ (3786300756998219980800 - 2236535390302235873 28\,\mathrm{i})a^7$$
$$- (1861552128281340026880 + 2068398340941735168000\,\mathrm{i})a^6$$
$$- (2446971090296433 63120 - 534019545039390350274\,\mathrm{i})a^5$$
$$- (2446971090296433 63120 + 534019545039390350274\,\mathrm{i})a^4$$
$$- (1861552128281340026880 - 2068398340941735168000\,\mathrm{i})a^3$$
$$+ (3786300756998219980800 + 2236535390302235873 28\,\mathrm{i})a^2$$
$$- (3690863685982361823120 + 1043642070788026200705\,\mathrm{i})a$$
$$+ (1329399368762685232320 + 2538165215672305923841\,\mathrm{i})\,.$$

## Result for $L = 4, M = 4$

$\hat{P}_4^{(4)}$

$$= -(2591467716551863855293739366 4573696 + 1888537223507387804182772927965 9653120\,\mathrm{i})a^{11}$$
$$+ (24630256608315177583960222249037046721 + 14791618858230018600341357060533136880\,\mathrm{i})a^{10}$$
$$- (35277224091134454445608489237612217220 - 10852527023685678060739087229387853360\,\mathrm{i})a^9$$
$$+ (15889997309911171224836659068505377795 - 29641482629099231105353029850095050160\,\mathrm{i})a^8$$
$$- (23546522614339811646300053852967111 60 - 26442493434928705349377006828321757760\,\mathrm{i})a^7$$
$$- (24084748945745632708905819811178781390 + 23883224392756331422017858 6885848160\,\mathrm{i})a^6$$
$$- (77789504038838948475406086177417062 40 + 25918642395117628948049529404717382240\,\mathrm{i})a^5$$
$$+ (49077616940393180206610546519782250190 + 29836741776308322341792887609402796640\,\mathrm{i})a^4$$
$$- (42996910595483003653184616761086396680 - 14211989812263351882467726396115543360\,\mathrm{i})a^3$$
$$+ (45532442407233332355320615831884779005 - 48519927643957199603125363113009532560\,\mathrm{i})a^2$$
$$+ (32886600144269651417496008844256049 96 + 41361215880861563790520315012683080880\,\mathrm{i})a$$
$$(-26525262588323420505310919969134672321 - 23569386823313100801648840241916702640\,\mathrm{i})\,,$$

$\hat{Q}_4^{(4)}$

$$= -(26525262588323420505310919969134672321 - 23569386823313100801648840241916702640\,\mathrm{i})a^{12}$$
$$+ (32627453372614465031966634907610 31300 - 60246588115935441832348044292342734000\,\mathrm{i})a^{11}$$
$$+ (70162699015548509932808380809218 25726 + 63311546502187218203466720173542669440\,\mathrm{i})a^{10}$$
$$- (78274134686617458098793105998698613900 + 33594627885776738217286391667276900 00\,\mathrm{i})a^9$$
$$+ (64967614250304351431447205588287627985 - 59478224405407553447145917459497846800\,\mathrm{i})a^8$$
$$- (10133602665317876012170614003038417400 - 52361135830046334297426536233039140000\,\mathrm{i})a^7$$
$$- 48169497891491265417811639622357562780\,a^6$$
$$- (10133602665317876012170614003038417400 + 52361135830046334297426536233039140000\,\mathrm{i})a^5$$
$$+ (64967614250304351431447205588287627985 + 59478224405407553447145917459497846800\,\mathrm{i})a^4$$
$$- (78274134686617458098793105998698613900 - 33594627885776738217286391667276900 00\,\mathrm{i})a^3$$
$$+ (70162699015548509932808380809218 25726 - 63311546502187218203466720173542669440\,\mathrm{i})a^2$$
$$+ (32627453372614465031966634907610 31300 + 60246588115935441832348044292342734000\,\mathrm{i})a$$
$$+ (-26525262588323420505310919969134672321 - 23569386823313100801648840241916702640\,\mathrm{i})\,.$$

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
