# Peer review of "Exact Spin Correlators of Integrable Quantum Circuits from Algebraic Geometry"

_SciPost Physics, doi:SciPost Phys. 19, 003 (2025)_

## Round 1 · Referee Report · Anonymous (Referee 1) · 2024-9-26

Strengths

1 - Demonstrates the applicability of computational algebraic geometry to a timely problem on correlation functions.
2 - Comprehensive and generally well-explained results.

Weaknesses

1 - The phenomenon noticed at the end of section 4 is not well explained.
2 - Section 5 would benefit from further physical discussion.

Report

The present work continues a series of papers, which were initiated by two of the authors and continued in various collaborations. The goal of those papers is to demonstrate how methods of computational algebraic geometry combined with rational Q-systems can serve to compute quantities in quantum integrable systems, where it is necessary to sum over all (admissible) eigenstates. After having studied partition functions of vertex and loop models in several papers, the collaboration here turns to two-point correlation functions in the six-vertex model of operators, that insert a domain-wall state of a given length in part of the systems.

The authors provide some salesmanship to justify that their computations are timely and relevant for applications in the theory of quantum circuits. The present reviewer believes that the results can stand for themselves, but also does not object to this positioning of the results.

Section 2 of the manuscript efficiently recalls the necessary background on integrability and defines the domain-wall operators. Section 3 contains some more advanced material, recalling the algebraic Bethe Ansatz and, crucially, how the overlap of the domain-wall operators with the on-shell Bethe states and the norm of the latter can be expressed as certain determinants. This brings together known ingredients in a user-friendly way and is well explained. After (34), instead of saying that the one-site shift operator is “not compatible” with most Bethe states, I think that the authors should rather say that it does not commute with the evolution operator (1), and go on to explain under which circumstances some Bethe states might still provide eigenvectors for it.

Section 3.2 drives home the point of applying computational algebraic geometry to this problem and recalls polynomial ideals, quotient rings, Gröbner bases and companion matrices. I feel that one point deserves some further discussion here. In the beginning of the section, the authors state that results for (48) and (51) cannot be obtained by direct numerical solution. Typical concrete results are given in section 4.1 and are polynomials in the six-vertex weights with integer coefficients. Once this polynomial property is known or assumed, why would it not be possible to obtain such results quite forwardly by computing (66) directly, using no knowledge of integrability whatsoever, but simply keeping track of the coefficients of the polynomials? This kind of question is all the more urgent since, at the end of section 3, the authors themselves admit using Lagrange interpolation to produce certain results. A similar claim is made again at the beginning of section 4.3.

After (57) the lack of an analytical formula for the Gaudin-like determinants in terms of $c_k$ is mentioned. I think this point is sufficiently important to be recalled in the conclusion section.

As already mentioned, results appear in section 4. The light-cone effect mentioned after (78) should be called by its proper name (as in the conclusion) and a more precise argument provided. In section 4.3 and 4.4 results are presented in the form of Lee-Yang zeros. Here the notion of condensation (accumulation point) should be precisely defined. Is the spread of zeros close to the real axis in figures 3 and 4 a real effect or an imperfection originating from the root solver? Finding roots of high-degree polynomials is a numerically challenging problem, so the authors should state which software is being used here. The notation in the first part of (81) should use the index j.

Section 4 ends with a point that the authors should really try to clear up before publication. Figure 6 shows that the condensation curves of $\tilde{\cal D}$ and ${\cal D}$ almost coincide in the lower-half plane, but differ substantially in the upper-half plane. The authors mention how these curves can be obtained (numerically) exactly by studying the equimodularity of eigenvalues, but seemingly only obtain approximations thereof by displaying the roots of high-degree polynomials. Are some of the eigenvalues of the two quantities {\em exactly} the same, or only approximately? If so, can the notable differences in the upper-half plane be explained by some of the eigenvalues incurring a vanishing prefactor, maybe by some symmetry argument?

Results in Fourier space are given in section 5. The choice of parameter values should be better explained (unless they just provide an arbitrary example), and if possible, this section would benefit from more effort in making a physical interpretation. For instance, are the spikes in figure 13 relevant for actual physical phenomena, and why do some of them converge but others seemingly not? Is the color coding of figure 13 the same as in figure 12?

Summarizing, this is a very nice paper, which is very likely to be publishable in SciPost Physics after the authors have addressed the above points.

Requested changes

The requested changes are integrated in the above report.

Recommendation

Ask for minor revision

  • validity: top
  • significance: high
  • originality: high
  • clarity: high
  • formatting: excellent
  • grammar: good

Author:  Yunfeng Jiang  on 2025-05-07  [id 5447]

(in reply to Report 1 on 2024-09-26)

See the detailed reply in the attached file.

Attachment:

Reply_to_referee.pdf

---

## Round 1 · Referee Report · Jules Lamers (Referee 2) · 2024-10-23

Strengths

  1. The authors give explicit results for certain correlation functions in integrable brickwork quantum circuits both in real and Fourier space for very small systems. Beyond that, further data is presented in the paper through numerous plots.
  2. In real space, the short and long (discrete) time asymptotics are discussed.

Weaknesses

  1. It is not quite clear to me what the physical content of the results (expressions and plots) are. Is the claim that these quantities should be particularly simple to compare to quantities measured in experimental platforms? What about the curves of zeroes; are they measurable? Or do they (or at least: are expected to) separate different phases, like for Lee--Yang? Or do the explicit expressions for low N serve more as a proof of concept, to show that the method gives very explicit results, and is the reader supposed to be able to similarly use the method (if they had the author's software package) to compute quantities that they are actually interested in. Similarly, are the values q=2 etc meant to be of particular physical relevance, or at least expected to be easily within the experimentally accessible range, or just for illustrative purposes? This should be clarified in the introduction, main text, and conclusion.

  2. It would be very helpful if the gap between the (somewhat vague or high-level) abstract description of the computational algebraic geometry and the concrete results can be closed by describing the actual computation (e.g. intermediate results) for at least one example to illustrate the theoretical story and give at least a flavour of the actual computations that are done

Report

The authors compute certain correlation functions for integrable brickwork quantum circuits with up to 20 qubits exactly using the algebraic Bethe ansatz and computational commutative algebra. Results in real and Fourier space are given explicitly for very small systems. In real space, short and long (discrete) time asymptotics are discussed. Beyond that, some further data is available on arXiv, and presented in the paper through various plots. The idea is that these results may be compared with (used to calibrate) experimental platforms for quantum simulation, which currently have such system sizes.

The derivations of these results appear to be correct. While I would not call this application of computational commutative algebra to brickwork quantum circuits groundbreaking, I agree that the methods may be useful. I cannot judge whether the resulting exact expressions and plots are indeed useful for comparison with such platforms.

The paper is fairly well written, although it would benefit from a careful re-reading to improve the grammar. The prerequisites from integrability are exposed in a relatively elementary way that I expect to be accessible for non-experts. The relevant commutative algebra is presumably harder to digest. In particular, there are no examples to illustrate how the computational methods that are described abstractly are used in practice to derive the results. This leaves somewhat of a gap between the theoretical part and the results that are presented.

Requested changes

See the two key points at 'Weaknesses'. Further feedback:

  • (10) and surrounding text: I believe tat \check{R} always has adjacent subscripts, in this case i,i+1 and i+1,i+2 rather than i,j etc
  • Remark 1: The phrase 'classical integrable' could be mistaken to mean 'classically integrable'; consider rephasing to clarify, and perhaps add that these lattice models are 2d
  • Following (18): clarify that $0<\gamma\ll 1$
  • Consider moving footnote 1 to p4 already
  • p9: mention that the resolution of the identity relies on the completeness of the Bethe ansatz, which will be discussed in Section 3.2
  • (38): here K is switched to N mid-formula. Pick one symbol and use throughout.
  • p11, before 'Rational Q-system': what does 'medium quantum numbers' mean - M=L, M=L/2, ...?
  • Rational Q-system: what does 'more rigorously' mean? which parts are/not rigorous?
  • Rational Q-system: it should be mentioned that completeness of the (Wronskian BAE =) QQ system has been proven for generic parameters by Mukhin--Tarasov--Varchenko, cf Chernyak--Leurent--Volin; I suppose this includes the case of alternating inhomogeneities \theta_j = (-1)^j u/2 as here (perhaps with a small twist)
  • above (53): \eta=0 is not the homogeneous case, cf text below (7); but \eta\to0 suitably interpreted (by rescaling u) is [to be corrected throughout the paper, eg also in footnote 3]
  • footnote 5: to clarify the scope of the method used here, it would be good to mention some key applications for which the original BAE have benefits
  • 'Symmetrization': what are "all the Q-polynomials"? Isn't there essentially only one? (In App C it becomes clear that there are several, but this is not mentioned in the main text, making the sentence unclear)
  • it is not explicitly stated what the 'Symmetrization procedure' precisely means. If it is the change of variables from the u_k to the c_k then I find it a strange name: the functions are symmetric to start with; they are just expressed more compactly in a basis of symmetric polynomials
  • the '(computational) algebraic geometry' used in the paper is really just '(computational) commutative algebra': no geometry is needed or used. I would suggest using the latter term, which sounds less scary, and might help making the paper look more accessible
  • before (60): 'All polynomials in $n$ variables ... form' is not very clear, -> 'The set [or: space] of [all] polynomials in $n$ variables ... forms'
  • top of p13: clarify that the coefficients a_i are themselves polynomials and generally not unique
  • is a 'basis of the ideal' just any generating set, or is there some condition that it should be minimal, e.g. one cannot remove any of the generators without changing the ideal
  • emphasise that the companion matrix P itself no longer depends on the x_i, but does still depend on u (and \eta for XXZ), as will be discussed below
  • p13, middle: make explicit which quantum numbers - L,M,Q_0
  • 'Dealing with free parameters': what is known about the form (or properties) of the resulting functions of u (and possibly \eta)? try to make this discussion more concrete. it will help understanding the type of interpolation that can be done
  • above and below (68): 'expect' vs 'find' - either prove, or make clear throughout that it is an expectation
  • (71): perhaps clarify that the polynomials are not necessarily homogeneous (of a fixed total degree) in b,c
  • (73) etc: to facilitate comparison, write polynomials in the same order - e.g. in the second line here, c^4 at the end
  • above 4.3: clarify in the final sentence that the result is initially independent of L
  • Fig 2: figure number '2' is missing. Caption: clarify whose zeros are shown
  • (81): \Prod_…^… ...^\otimes is very strange notation. Either \bigotimes_…^… ... or ...^{\otimes L}
  • should 'cylinder partition function' be 'torus partition function'? or is it meant to be on an infinite cylinder or with fixed boundaries at the ends?
  • clarify that Z_{L,n}(u) is a polynomial only up to an overall factor of u^… (in your normalization of the weights)
  • Fig 5: should this be \widetilde{D}_4(500)?
  • Fig (4,5,)6: pick colours that differ more
  • in the caption mention which colour is which function (or add a legend)
  • perhaps mention that blue has extra branches in the lower half plane close to the real intervals \pm[.5,.9]
  • before Sect 5: clarify whether this argument also explains why the shapes are similar (rather than just the same endpoints for the curves), and whether the curves have a physical meaning - do they also separate different phases?
  • I'd say that Q_0, rather than e^{i Q_0}, is the (quasi-)momentum
  • the quotient ring is smaller: than what - if we choose another fixed momentum, or when we do not fix the momentum
  • below (87): can we easily see why here we need \xi rather than x=\xi^2 like before?
  • above (91): to clarify the scope of the method used here, explain that \xi=9 and q=2 (integers) is much faster than [rational numbers? or only compared to real/complex numbers?]
  • p22 onwards: explain the meaning of the \sim (range of integers; at the bottom of p25 a dash is used once as well, is there a difference?)
  • Sect 6: compute the correlation -> compute certain correlation; or is the claim that from these special correlation functions one can obtain all others?
  • recall that \xi = e^u and q=e^\eta to help remind the reader
  • before (99): clarify what 'symmetric property' here means - does one take the complex conjugate of the scalar product under the assumption that it is real (and using $B^\dag$ = $C$)?
  • before (107): clarify that this computation depends on a (Gröbner basis and thus) choice of generators ${g_j}$ and order
  • App B.4: which algebraic extension - of scalars?
  • Does this mean that q=11/10 would take rather longer than q=2? If so, mention this in the main text.
  • Before App C: explain whether this interpolation relies on some knowledge about the form or properties of the coefficient functions
  • Young diagram: (M,L-M) should be (L-M,M) to be a partition. add a figure to show which alignment of the boxes you have in mind and explain terms such as 'lower left'
  • (113): recall that q=e^\eta
  • below (113): clarify what 'up to proportionality' means - by a constant? what can it depend on? Surely the degrees on the two sides of (112) must match, so one cannot just freely rescale the different Q-functions separately
  • Zero remainder condition: what does 'the upper conditions' mean?
  • before (118): introduce both a new variable w and an equation for that variable

Typos and grammar

p1 for calibration -> for the calibration we use algebraic -> we use the algebraic geometry to obtain -> geometry, to obtain long time limit -> long time limits

p2 researches have -> research has

p3 We found that, for -> We find that for independent from -> independent of [to be corrected throughout the paper] Fourier space 5. -> Fourier space.

p4 factorize out $\sigma^z$ operator -> [e.g.] factor out an operator built from $\sigma^z$

p5 was use by Destri -> was used by Destri chain and further -> chain and was further want to emphasis -> want to emphasize are interested in the current -> are interested in in [/for] the current

p6 physics the research -> physics, research site-$k$ -> site $k$ remove the redundant space starting the line after (18) goal is computing -> goal is to compute

p7 compute time evolution $\mathcal{U}$ -> [e.g.] compute the time evolution by $\mathcal{U}$

p8 qubits towards right -> qubits towards the right

p9 resolution of identity -> resolution of the identity

p10 Defining the following -> Define the following

p11 perform different analysis -> [clarify: either] perform a different analysis [compared to what? or] perform various analyses Numerical solution cannot -> Numerical solutions cannot the sum over solutions of BAE in (48) and (51) -> the sum in (48) and (51) over solutions of the BAE various magnon numbers -> [clarify: either] any magnon number [or] several magnon numbers [but not all?] problem of Bethe -> problem of the Bethe apply the method of rational -> use the rational in rational Q-system -> in the rational Q-system Q-polynomials for -> Q-polynomial for 0$) as -> 0$) are defined as For XXZ chain -> For the XXZ chain We compare efficiency -> We compare the efficiency

p12 of rational Q-system -> of the rational Q-system either solve these ... manipulate -> either solves these ... manipulates e.g. explicit -> e.g. the explicit Groebner vs Gröbner: choose one way to format and use throughout Appendix D vs appendix B: choose one way to format and use throughout introduce the abbreviation 'AG' used below (or spell it out there)

p13 fact of ideal -> fact of the ideal different set of -> different sets of kind of bases -> kind of basis canonical basis $e_j$ -> canonical basis ${e_j}$ [and similarly below] called \emph{companion -> called the \emph{companion computation of Gröbner -> computation of the Gröbner certain computation for -> certain computations for computation of companion -> computations of the companion Now coming -> Now we come crucial results -> crucial result is that, the sum -> is that the sum

p14 definition of \mathcal{U} -> [e.g.] definition of the discrete two-step time evolution operator \mathcal{U} [to help remind the reader] where $b(u)$ and $c(u)$ -> where $b(u) = ...$ and $c(u) = ...$ [give functions in terms of \varphi to help remind the reader] using AG computation -> [e.g.] using the AG computation plug ... to -> plug ... into [and similarly elsewhere]

p15 (73) etc: check that equations end in the correct punctuation ( . or , ) [throughout, all the way to the appendices] we expect for small -> we expect that for small

p16 obatined Lee-Yang -> Lee--Yang [throughout, also for Beraha--Kahane--Weiss, Fabricius--McCoy; cf the single name Frühbis-Krüger]

p17 phase structure here -> phase structure. Here not exhibit clear -> not exhibit a clear the rest two -> the other [or] remaining two [and similarly elsewhere]

p18 limit correspond to -> limit corresponds to move footnote 7 to after the . to avoid it looking like a power

p19 aslo applies o

p20 as eigenvalues -> as the eigenvalues

p21 we call ... as the ... -> we call ... the ... [and similarly elsewhere] . we find -> . We find few more results -> few more results for

p22 of arXiv -> of the arXiv

p23 complicated as expected -> complicated, as expected infinite length $L -> infinite length limit $L the similar observation -> a similar observation than the 2-magnon -> than for the 2-magnon

p24 on Bethe ansatz -> on the Bethe ansatz of BAE analytically -> of the BAE analytically

p25 to partial -> to a partial which is a rational function -> which are rational functions

p26 . the curves -> . The curves where $\alpha,\phi$ being -> where $\alpha,\phi$ are results for quickly -> results quickly of $\omega$ have poles -> of $\omega$ has poles set small -> set a small for finite $L$ result -> for a finite $L$ result to the domain wall -> to domain wall physics, physical -> physics, but that physical see e.g -> see e.g.\ such result -> such results

p27

p27 promising -> possible

p29 Noether -> Noetherian [?] finitely generate -> finitely generated is vanishing -> vanishes

p30 set of basis -> basis [?] Such convenient basis indeed exists and are called -> Such a convenient basis does indeed exist and is called a the for any monomial -> then for any monomial definition of Gröbner -> definition of a Gröbner division of ... towards ... -> division of ... by ... $t$ -> $r$

p31 With Gröbner -> With a [or] the Gröbner by any elements -> by any element with respect to Gröbner -> with respect to the Gröbner computational time -> computation time The latter computation is -> The computation of the companion matrix is [clarifies 'latter']

p32 for XXZ spin -> for the XXZ spin At each node ... associate with ... -> To each node ... associate ... For XXZ spin -> For the XXZ spin as a function -> as functions left boundary is -> left boundary are

p33 In general, we have -> In general, we consider algebraic equation involving -> algebraic equations involving of these set -> of this set that involve only -> that involves only [fix the grammar of the two sentences] By solving ... (119) ... and obtain $w=\widetilde{S}({c_k})$.

p34 a) -> (a)

p43

De Vega -> de Vega [?]

Recommendation

Ask for minor revision

---

## Round 2 · Referee Report · Jules Lamers (Referee 2) · 2025-5-27

Report
Recommendation
Publish (meets expectations and criteria for this Journal)

---

## Round 2 · Author Response

List of changes

---

## Round 2 · List of Changes



---

## Editorial Decision

published